# Taste Sensor Assessment of Bitterness in Medicines: Overview and Recent Topics

**DOI:** 10.3390/s24154799

**Published:** 2024-07-24

**Authors:** Takahiro Uchida

**Affiliations:** 1Food and Health Innovation Center, Nakamura Gakuen University, 5-7-1, Befu, Jonan-ku, Fukuoka 814-0198, Japan; ttakahiro@nakamura-u.ac.jp; 2Faculty of Pharmaceutical Science, Mukogawa Women’s University, 11-68, Koshien 9-Bancho, Nishinomiya 663-8179, Japan

**Keywords:** taste sensor, electronic tongue, bitterness, human taste receptor, allostery, adherence

## Abstract

In recent decades, taste sensors have been increasingly utilized to assess the taste of oral medicines, particularly focusing on bitterness, a major obstacle to patient acceptance and adherence. This objective and safe method holds promise for enhancing the development of patient-friendly medicines in pharmaceutical companies. This review article introduces its application in measuring the intensity of bitterness in medicine, confirming the achievement of taste masking, distinguishing taste differences between branded and generic medicines, and identifying substances to suppress bitterness in target medicines. Another application of the sensor is to predict a significant increase in bitterness when medicine is taken with certain foods/beverages or concomitant medication. Additionally, to verify the sensor’s predictability, a significant correlation has been demonstrated between the output of a bitter-sensitive sensor designed for drug bitterness (BT0) and the bitterness responses of the human taste receptor hT2R14 from BitterDB (huji.ac.il). As a recent advancement, a novel taste sensor equipped with lipid/polymer membranes modified by 3-Br-2,6-dihydroxybenzoic acid (2,6-DHBA), based on the concept of allostery, is introduced. This sensor successfully predicts the bitterness of non-charged pharmaceuticals with xanthine skeletons, such as caffeine or related compounds. Finally, the future prospects of taste sensors are discussed.

## 1. Introduction

Bitter-tasting medicines often lead to poor acceptance or adherence among patients and, in the worst cases, may result in outright rejection [1,2]. To develop well-taste-masked formulations, it is essential to initially quantify the bitterness intensity of each active pharmaceutical ingredient (API) and then determine preparative conditions that achieve adequate bitterness reduction while maintaining good palatability.

Human sensory testing has traditionally been a primary method for determining the bitterness intensity of oral pharmaceuticals and remains an available option [3,4,5,6]. However, in recent years, the use of human sensory tests has become ethically problematic in the development of various oral medicines, including oral anticancer drugs with high toxicity. Moreover, human sensory tests may be influenced by factors such as individual anatomy, psychophysics, sex differences, illness, or dietary habits [7,8,9,10,11]. Conducting sensory tests accurately and carefully is essential, including strict prohibitions on consuming food or drinks during the test, which can be burdensome for participants. Additionally, it is crucial to note that molecules not approved by the Food and Drug Administration (FDA) cannot be tested using these methods, as mentioned in the previous article [12]. In addition to human sensory testing, alternative methods utilizing taste bud cells extracted from frogs [13,14], small mammals [15,16], and cells expressing bitter taste receptors have been explored [17,18,19]. However, these experiments require a certain level of expertise and entail costs. Moreover, experiments involving animals raise ethical concerns from an animal welfare perspective.

The use of artificial taste sensing systems capable of evaluating the bitterness of medicines is gaining traction as a means to replace human sensory tests and studies involving animals, owing to their reproducibility, ethical considerations, and the reduction of associated burdens. For bitterness evaluation of medicines, various taste-sensing systems—also known as electronic tongues, i.e., e-tongues—have been employed [20]. These systems utilize different types of sensors, including electrochemical [21,22,23], optical [24,25], or enzymatic sensors (biosensors) [26,27].

The taste sensor, initially developed by Toko, features an electric tongue equipped with multi-array lipid/polymer membranes on the sensor electrodes [28]. This sensor was engineered to discern and characterize five basic tastes as well as astringency by detecting changes in membrane potential induced by taste substances [29,30,31]. The first assessment of bitterness in medicine using this multichannel sensor with global selectivity was conducted by Toko and co-workers, focusing on quinine hydrochloride, a representative alkaloid with a bitter taste and a therapeutic agent for malaria parasites [32].

Since 2000, Uchida and co-workers have been pioneers in the application of multichannel taste sensors for qualitative and quantitative assessment across a wide range of oral medicines, including nutritional medicines [33,34,35], and demonstrated its usability in predicting bitterness of medicines when co-administered with certain beverages, foods, or other medications. [36,37,38]

In addition to Toko’s taste sensor system, the Astree sensor system has also been utilized in pharmaceutical development, employing multiple probes that are not specific to the five tastes. This system has been employed for pattern analysis in evaluating bitterness suppression by artificial sweeteners [39] and in optimizing taste-masking formulations [40,41].

In a 2006 review article, Breitkreutz and Boos highlighted the challenges encountered by pediatric patients and the elderly when swallowing solid dosage forms intended for oral use [42]. They pointed out that both patient demographics prefer smaller-sized particulates or liquid dosage forms over traditional tablets or capsules. Furthermore, they suggested that the new EU registration requirements for the development of pediatric medications would stimulate research into intelligent oral drug formulations tailored for these specific populations. The taste sensor target has to cover new dosage forms, such as orally disintegrating tablets (ODTs) [43], orodispersible films (ODFs) [44,45], minitablets [46,47], and formulations utilizing hot-melt extrusion technology that have entered the market as patient-friendly medications [48,49]. Currently, there is a trend in the pharmaceutical industry towards formulations featuring a “universal technique” or “universal design” with global palatability [50].

The establishment of the BitterDB website has provided access to more information on bitter drugs and bitter taste receptors, with this database information being adopted in many research papers [51,52] and proving to be useful for better understanding bitterness perception and taste masking. Regarding human bitterness receptors hT2Rs, 25 subtypes have been reported [53,54,55]. Among these subtypes, hT2R14 has been identified as a representative and significant bitter receptor, with extensive studies on its structure and function [56,57]. Docking simulations have also yielded valuable insights [57]. 

Haraguchi et al. have elucidated that the Insent multichannel taste sensor designed specifically for drug bitterness exhibits high sensitivity to bitter sensors. Furthermore, with regard to drugs that act as agonists of hT2R14, a significant correlation between the output of the bitter-sensitive sensor (BT0) and the corresponding responsiveness of hT2R14 withdrawn from the Bitter database has been confirmed [58,59]. This knowledge holds promise for verifying the predictability validity of the Insent taste sensor.

Significant bitterness-suppressing effects of chlorogenic acid [60] and umami peptides were reported recently [61]. These substances are non-toxic and readily available.

Allostery is a phenomenon observed in numerous enzymes and receptors, including taste receptors. It describes how the binding of a ligand at one site can affect the binding of a different or similar type of ligand at another distant site of the enzyme or receptor molecule [62,63,64]. 

Toko and co-workers developed a novel taste sensor based on the concept of allostery and demonstrated its application, shedding light on its mechanism [65,66,67]. The new sensor electrode was prepared from a membrane comprising the lipid tetradodecylammonium bromide (TDAB), the plasticizer dioctyl phenylphosphonate (DOPP), and polyvinyl chloride (PVC), which was then modified by immersion in a solution containing 0.05 wt % of 2,6-dihydroxybenzoic acid (2,6-DHBA). This sensor significantly enhanced the output of non-charged bitter xanthine derivatives compared to what was obtained using the conventional bitter-sensitive sensor, BT0. This enhancement in the new sensor’s performance is expected to arise from the interaction between caffeine or related compounds and 2,6-DHBA, facilitated by the formation of hydrogen bonds between the hydroxy group of hydroxybenzoic acid (HBA) and the carbonyl group or N (imidazole) of caffeine (or related compounds), along with the π―π interaction between the aromatic rings. 

## 2. Bitterness Assessment in Medicines by Taste Sensor

The Insent multichannel taste sensor, known as an e-tongue system, provides significant benefits for evaluating five tastes and astringency in various substances, including foods, beverages, and pharmaceuticals [31,68,69]. In those substances, it is particularly effective in assessing the bitterness of a wide range of medications. Especially, the e-tongue is good for assessing bitterness in a wide range of medicines and can predict the enhancement of medicines with certain drinks/foods or other medicines.

The Astree sensor system, another e-tongue, has also been utilized for a wide range of substances. Even in pharmaceutical fields, systems using sensor arrays not specific to the five tastes have been employed for measurement and pattern analysis to evaluate bitterness suppression in formulation design [39]. They have been used to optimize the type or amount of polymer/sweetener in taste-masking formulations [40,41]. 

At the end of this chapter, the author introduces the characteristics of Insent and Astree e-tongues and summarizes their differences, including advantages and/or tasks in applications to pharmaceuticals.

### 2.1. Bitterness Assessment in Medicines by Multichannel Taste Sensor

As described in the introduction, up until 2000, there was a lack of systematic research on the bitterness of many oral medicines, with the exception of studies on quinine conducted by Takagi et al. [32]. The study demonstrated that quinine exhibited high responsiveness in the bitter-sensitive membrane while eliciting no responses in the other four sensors sensitive to sourness, saltiness, sweetness, and umami. Furthermore, the suppression of bitterness in quinine by sucrose was quantified accurately using the bitter-sensitive sensor. Notably, this sensor measurement is fully automated and can evaluate the bitterness in drugs or substances with high accuracy and reproducibility [32].

In 2000, Uchida and co-workers assessed the bitterness of 11 commercial medicines using a multichannel taste sensor (402B) with sensor membranes [33]. This study compared the bitterness of these medicines as assessed by the multichannel taste sensor with human gustatory sensation tests involving 15 volunteers. The lipid components of the taste-sensor membrane and the measuring procedure are shown in Figure 1 and Figure 2, respectively.

One set of measurements, consisting of five repetitions for a single sample, will be completed in 20 min. In this study, the relative sensor output (R) was adopted as the explanatory variable. For basic drugs like quinine and trimebutine, which contain a cationic amino group in their molecules, there was a relatively large relative response electric potential (mV) in channels 1 or 2, which contain negatively charged membranes. These drugs also exhibited considerable bitterness in human gustatory sensation tests. However, anionic acidic drugs such as diclofenac sodium and salicylic acid, which contain an anionic carboxyl group in their molecules, showed a negative electric potential in channels 5 or 6, which contain positively charged membranes. This information proved useful even though these drugs taste sour rather than bitter. For pharmaceuticals like theophylline and metronidazole, which are neutral, and caffeine, which is a weak basic drug, the relative response electric potential (mV) of channels containing negatively charged membranes did not increase. It is interesting to note that despite the absence of dissociation within the molecule and no charge, certain substances still exhibit considerable and severe bitterness in human gustatory sensation tests. This phenomenon has posed a challenge that remained unresolved until recently, as discussed in the final section.

Secondly, Uchida et al. focused on assessing the bitterness of 10 basic medicines (all hydrochloride salts) with bitter tastes and different structures, including amitriptyline, d-chlorpheniramine, dextromethorphan, dibucaine, diltiazem, imipramine, promethazine, propranolol, trimebutine, and quinine, using a multichannel taste sensor [34]. In this study, a new measurement protocol called semi-continuous measurement was adopted, as shown in Figure 3. Measuring one sample five times as one set will be completed in 30 min. As explanatory variables, sensor output (S), the change of membrane potential caused by adsorption (C), attenuation of C for up to 150 s after initial exposure (C’), and C/S were used to predict the bitterness score of the medicines in multiple regression analysis.

Eleven volunteers took part in human gustatory sensation tests. Multiple regression analysis yielded a correlation coefficient (r) of 0.824 for the bitterness score predicted by the taste sensor using C’ for channel 2 and C/S for channel 4, as depicted in Figure 4, compared to using C/S for both channels 2 and 4 (r = 0.734) [34]. This approach and the results obtained in the study seem to provide reliable predictability for assessing the bitterness of fundamental medications, even in the presence of substantial variances in pharmaceutical structures.

In general, certain oral antibiotics are notorious for their poor taste, often resulting in low adherence rates among patients, including infants [70,71,72]. To tackle this issue, Uchida et al. assessed the bitterness of nine commercial antibiotics (clarithromycin, erythromycin, cefdinir, doxycycline, vancomycin, tetracycline, minocycline, oxytetracycline, and bacampicillin) using a multichannel taste sensor. Human sensory tests were also conducted with nine volunteers [35]. The bitterness of 0.1–0.3 mM solutions (except for clarithromycin, which was tested as a suspension) of the antibiotics was employed for sensor measurements. Three variables were utilized to estimate bitterness in single or multiple regression analyses, and principal component analysis was carried out as described in the preceding paragraph. Importantly, a strong correlation was observed between the obtained bitterness scores and the predicted scores using C from channel 2 of the sensor (r^2^ = 0.870, *p* < 0.005) and C/R values for channels 2 and 3 (r^2^ = 0.947, *p* < 0.005). Even with single regression, a robust relationship between the predicted taste sensor and the obtained bitterness intensity in humans was affirmed. Clarithromycin (CAM) displays low aqueous solubility but emerges as the most bitter among the nine antibiotics, based on sensory data. In pharmaceutical research and development, various approaches to mask bitterness have been explored, including physical methods [73,74,75], chemical methods [76,77,78], and organoleptic methods [79,80,81], either individually or in combination [82,83]. Among these methods, physical methods such as coating are typical and conventional masking techniques. Clarithromycin (CAM), known for its severe bitterness, led to the production of commercial clarithromycin dry syrup (CAMD) (Clarith dry syrup, Taisho Pharmaceutical Co. Ltd., Tokyo, Japan) using a physical masking method involving polymers [84,85,86]. In their investigation, Uchida et al. assessed the bitterness of CAMD using taste sensors. The bitterness of the dry syrup was estimated to be reduced to about 1% of that of an equivalent powder suspension [35]. In other words, nearly 99% of the bitter taste was successfully masked in the dry syrup formulation, as demonstrated not only by human gustatory sensation but also by taste sensor prediction. Thus, the multichannel taste sensor is applicable to dry syrup suspensions with different concentrations and can successfully predict bitterness quantitatively. 

Kobayashi et al. introduced a new type of sensor, the TS-5000Z [87]. Their article introduced the new bitterness taste sensor (BT0) and astringency taste sensor (AE1), demonstrating that sensor outputs obtained by these sensors exhibit a high correlation with human sensory scores. This e-tongue and its attached sensors were used to assess the tastes of various substances and for their quality control.

### 2.2. Prediction of Bitterness Enhancement in Bitter Medicines with Foods, Beverages, or Other Medications by Multichannel Taste Sensor

Tanigake et al. conducted multichannel taste sensor assessments and human gustatory sensation tests for mixed samples comprising 1 g of commercial clarithromycin dry syrup formulation (branded as Clarith^®^ dry syrup) with 25 mL of various liquids, including water, coffee, tea, green tea, cocoa, milk, and a sports drink [36]. Significantly, a robust correlation was observed between the results from human sensory tests and the predicted values calculated based on multiple regression analysis using data from channels 3 and 4, as depicted in Figure 5, and the ratio of data from channels 3 and 4 of the taste sensor (r^2^ = 0.963, *p* < 0.005). Human sensory tests were conducted using the equivalent density examination method developed by Katsuragi [88]. The bitterness score shown on the horizontal axis in Figure 5 represents human bitterness scores ranging from 0 to 4—0 representing very weak bitterness and 4 indicating severe bitterness intolerable for more than 10 s. Clarithromycin dry syrup (CAMD) was designed to mask the drug at a pH of 6.8, which is similar to the pH conditions of the oral cavity. However, under gastric conditions with acidic pH, CAMD is formulated to ensure rapid and complete release of clarithromycin, thereby achieving optimal bioavailability. This balance between adequate taste masking and maintaining full bioavailability represents a crucial trade-off in the development of medicines with a bitter taste, as reported in a previous article [89].

In the realm of pediatric medicine with bitter taste, several research papers have explored the effect of foods or beverages using taste sensors.

For instance, Ishizaka and co-workers assessed the bitterness of 18 different antibiotic and antiviral drug formulations widely used to treat infectious diseases in children and infants [37]. The use of a multichannel taste sensor facilitated the anticipation of bitterness amplification caused by acidic sports drinks in three macrolide antibiotic formulations (ERYTHROCIN^®^ dry syrup), clarithromycin (CLARITH^®^ dry syrup for pediatric use), and azithromycin (ZITHROMAC^®^ fine granules for pediatric use as suspension sample). Conversely, a decrease in bitterness intensity was foreseen for an acidic sports drink suspension of an amantadine product (SYMMETREL^®^ fine granules) compared to an aqueous suspension, as illustrated in Figure 6. In the figure, concerning Norfloxacin and SMZ/TMP, the pH of each medicine with and without an acidic sports drink is not significantly different; therefore, sensor output enhancement might not occur. In contrast, in the case of Cefcapene, a moderate increase in pH with an acidic sports drink led to an enlargement in sensor output. Hence, the taste sensor could precisely forecast the escalation in or mitigation of bitterness intensity in macrolide or other varieties of antibiotic formulations with or without acidic sports drinks. 

Ishizaka also identified a medicine that significantly increased the bitterness of clarithromycin dry syrup (CAMD) when given together and suggested a method to mitigate this heightened bitterness [38]. In that study, the intensified bitterness was assessed through taste tests, pH measurements, and multichannel taste sensor analysis in a combined sample of CAMD (a basic drug) and L-carbocysteine dry syrup (an acidic drug), both used concurrently for upper respiratory tract inflammation and bronchitis, as indicated in the package insert. A notable increase in bitterness was noted when CAMD was combined with the acidic L-carbocysteine powder, likely attributed to the low pH (3.40) of the suspension created from these two formulations.

This acidity probably facilitated the heightened release of clarithromycin due to the dissolution of the alkaline polymer film coating. Similar outcomes were observed for the bitterness increase in basic macrolide azithromycin fine granules when administered concurrently with L-carbocysteine. Since the pH of L-carbocysteine alone is approximately 2.94, the addition of L-carbocysteine to azithromycin fine granules creates acidic conditions, thereby facilitating the rapid release of azithromycin, which is a basic drug. Interestingly, it was discovered that chocolate jelly available in the Japanese market, with a neutral pH, effectively masked the bitter taste of macrolide drug formulations when administered alongside acidic drug formulations in patients [3].

Thus, the multichannel taste sensor can predict that the concomitant administration of CAMD or other macrolides (basic drugs) with L-carbocysteine (acidic drug) dramatically enhances the bitterness of macrolides. Consequently, medical staff in hospitals can recommend to patients or their guardians to take the two drugs separately—first taking L-carbocysteine and then, 20 min later, taking CAMD—as a highly recommended approach [90,91].

### 2.3. Assessment of Taste Masking in Medicines by Astree Sensor System

In comparison with the clarithromycin study in the previous section, J Nawaf Abu-Khalaf et al. also assessed the bitterness of commercially available clarithromycin pharmaceutical suspensions (the brand Klacid^®^ and two generics, K1 and K2) in the Palestinian market using the Astree sensor system [92]. The assessed preparations showed significantly different tastes in the order of Klacid^®^ > K1 > K2, as suggested by the electric tongue and confirmed by in vivo results from pediatric patients using a panel test. The ranking of the three products was determined by volunteering community pharmacists. The authors concluded in the article that the system is useful for ranking bitterness among branded and generic products.

Another taste-masking evaluation using the Astree sensor system was conducted by Kayumba et al., as follows: To formulate a taste-masked version of quinine sulfate suitable for pediatric use, pellets were prepared through extrusion–spheronization [93]. Eudragit EPO was used as a coating agent to mask bitterness. However, the addition of 15% dibutyl sebacate as a plasticizer resulted in pellet agglomeration under storage conditions of 40 degrees Celsius and 75% relative humidity. In contrast, stearic acid at the same concentration reduced pellet sticking. Dissolution testing indicated that coatings of 10%, 20%, and 30% Eudragit EPO released 9.2%, 5.9%, and 2.1% of the drug dose within the first 5 min, respectively, correlating with bitterness scores obtained from the Astree sensor system. A 20% Eudragit EPO coating was deemed optimal for forming a uniform film and sufficiently delaying the release of quinine sulfate to mask bitterness after administration. An immediate release occurred under acidic conditions, highlighting the utility of the Astree e-tongue in optimizing the amount of polymer coating.

Li et al. implemented a unique modification of the simplex design to optimize bitterness masking in a formulation using the Astree e-tongue. Three variables—concentrations of taste-masking polymers (Amberlite and Carbopol) and pH of the granulating fluid—were evaluated [94]. The response measured was bitterness distance, assessed via e-tongue with principal component analysis, indicating formulation efficiency. Contour plots and polynomial equations of bitterness distance were generated based on formulation composition and pH. Interactions between the polymers and pH were observed to reduce bitterness, influenced by the pH-dependent ionization and complexation properties of the polymers, which affect drug solubility and bitterness perception. Optimal taste masking occurred at pH 4.9 with an Amberlite/Carbopol ratio of 1.4:1 (*w*/*w*), correlating with human gustatory sensation studies. Thus, the use of a modified simplex experimental design with e-tongue response proved effective for optimizing taste masking using ionic binding polymers.

Liu et al. utilized the Astree e-tongue to predict the bitterness of berberine hydrochloride. They developed a bitterness prediction model (BPM) integrating taste panel evaluations, e-tongue data, and a genetic algorithm-back-propagation neural network (GA-BP) [95]. The performance of the GA-BP model was compared with multiple linear regression, partial least squares regression, and BP methods. The BPM achieved a determination coefficient of 0.99965 ± 0.00004, root mean square error of cross-validation of 0.1398 ± 0.0488, and cross-validation correlation coefficient of 0.9959 ± 0.0027, surpassing the other models. This model accurately predicts the bitterness of berberine hydrochloride across concentrations and serves as a reference for developing similar models for other drugs. The study’s algorithm enables rapid and precise quantitative analysis of e-tongue data.

Although not directly related to pharmaceuticals, Kovacs et al. conducted a quality control study using an Astree e-tongue to evaluate sensor signal disturbances and develop drift correction techniques. They tested apple juice samples at various temperatures and measured pH changes to assess cross-contamination. Different sequences of model solutions and apple juice were used to evaluate memory effects. Over six weeks, they monitored model solutions representing basic tastes and commercial apple juice to model sensor signal drift. Their findings indicated that temperature, cross-contamination, and memory effects affected sensor signals [96]. They proposed three drift correction methods expected to be applicable for long-term e-tongue measurements in pharmaceutical quality control.

The Astree sensor system, which uses a sensor array, is not specific to five specific tastes. Despite this, it is utilized for measurement and pattern analysis in evaluating bitterness suppression in formulation design and for optimizing the type or amount of polymer/sweetener in taste-masking formulations. The taste sensor is expected to be applicable for long-term measurements of samples and suitable for quality control in pharmaceuticals.

### 2.4. Characteristics of Insent Taste Sensor and Astree Sensor System in Pharmaceutical Applications 

There are distinctive features in the two taste sensors, making it difficult to definitively compare the superiority of one over the other. As mentioned above, both taste sensors contribute to pharmaceutical development using various methods. 

As stated at the beginning of this chapter, I will briefly summarize the characteristics of the two representative e-tongues in the next section. I will also describe the performance qualifications of these sensors in the upcoming Section 3.4.

#### 2.4.1. Insent Sensor

##### Overview

The Insent multichannel taste sensor, known as an e-tongue system, provides significant benefits for evaluating five tastes and astringency in various substances, including foods, beverages, and pharmaceuticals, and has been used worldwide.

##### Advantages

The Insent taste sensor can quantify the bitterness intensity of a wide range of pharmaceuticals, from oral medications to traditional Chinese medicines, as well as the degree of bitterness masking.The assessment is achieved using a lineup of bitterness-specific sensor membranes and astringency-specific membranes.It can predict an increase in bitterness when co-administered with certain foods/drinks or other pharmaceuticals.It can quantitatively assess bitterness at the moment pharmaceuticals are taken into the mouth and also assess the bitterness that lingers as an aftertaste, even after rinsing following ingestion.It can also quantify the bitterness intensity of pharmaceuticals and formulations in a suspended state.

##### Challenge 

Sensor output of non-charged substances

The sensor output of non-charged substances, such as caffeine and related compounds like theophylline or theobromine, has been very low previously. Nevertheless, as described in Section 7, Toko and co-workers have developed a new sensor that can recognize the caffeine scaffold.

##### Duration

Taste sensors made of lipid membranes must be handled carefully and require a preconditioning procedure before actual measurement. Nowadays, the durability of lipid membranes has been significantly improved.

#### 2.4.2. Astree Sensor

##### Overview

The Astree electronic taste system enables representation through pattern analysis from multiple sensors and utilizes the selectivity of individual sensors. It supports a diverse range of applications, including benchmarking and mapping, quantitative taste assessment using standard substance addition methods, and constructing correlation models with sensory evaluation to assess masking effects.

##### Advantages

It is possible to objectively determine that formulations with smaller Euclidean distances between active drugs and placebos have weaker original drug flavors, meaning they exhibit stronger masking effects.It is also possible to compare between branded drugs and generic medicines.Utilizing sensors that respond to high-molecular-weight ingredients or sweeteners in formulation raw materials allows for the design of formulations that mask bitterness.Long durability.

##### Challenge

Each sensor array’s specificity to five tastes might not be guaranteed; therefore, it seems difficult to determine the bitterness of the drug itself quantitatively, and an appropriate benchmark will be needed. No information about the astringency evaluation of the drug involved is available. 

Regarding both e-tongues, a more compact, smaller size sensor might be needed in the future.

## 3. Advanced Taste Sensor Application to New Dosage Forms

ODTs have emerged as dosage forms originating in Japan and are not only utilized by elderly individuals with swallowing difficulties but are also accessible to patients across various age groups, making them a dosage form with universal design characteristics [97]. Presently, some ODTs are designed to incorporate small-size functional particles for taste masking when the involved drug shows severe bitterness [98,99,100].

There has been a development of new taste sensor membrane lineups, as observed in the Insent taste sensor application. Additionally, advanced analytical methods have been adopted in the Astree taste sensor system. These updates in taste sensors seem to contribute to new formulation designs and the choice of appropriate additives. Of course, taste masking and its assessment of conventional dosage forms, such as liquid formulations, must not be forgotten.

Particularly concerning orally disintegrating formulations such as ODTs, there is a need to assess the time course of bitterness under oral cavity conditions and to consider the timing of formulation disintegration and subsequent drug dissolution. Therefore, the combination of a taste sensor, such as an e-tongue, and an apparatus capable of predicting formulation degradation/dissolution under oral cavity conditions is desirable, as explained in the following section.

### 3.1. Assessment of Taste Masking in ODTs by Multichannel Taste Sensors

This review article also focused on the application of taste sensors to new dosage forms that rapidly disintegrate and dissolve bitter medicines within the oral cavity. Concurrently, advancements are being made in the development of new sensor membrane lineups for Insent multichannel sensors [87,101], and sophisticated analytical methods for the α-Astree system are underway [102,103]. Moreover, numerous articles concerning bitterness transduction, including studies on bitterness receptors, have been published [104,105,106].

Harada et al. demonstrated the usefulness and broad applicability of a multichannel taste sensor (402B Insent sensor) in conjunction with a novel disintegration testing apparatus, ODT-101, in the development and evaluation of the taste of orally disintegrating tablets (ODTs) [107]. The ODT-101 apparatus can simulate ODT disintegration in the oral cavity. The authors screened masking agents for their ability to suppress the bitterness of propiverine hydrochloride and produced ODTs of propiverine hydrochloride with various masking agents. The tastes of these ODTs were then assessed sequentially by integrating the taste sensor with the new disintegration testing apparatus, ODT-101, to replicate conditions in the oral cavity. Consequently, the taste of propiverine hydrochloride and the efficacy of various masking agents in ODTs were assessed using the combination of the taste sensor and ODT-101, yielding results consistent with human gustatory sensation testing. This underscores the utility of this novel application. 

Additionally, Harada also demonstrated the utility of the multichannel sensor in screening bitterness agents against propiverine as a model drug [108]. 

The special measurement method involves placing an ODT sample on a stainless-steel porous plate with a weighted shaft capable of vertical and rotational movement. A pump automatically adjusts 450 mL of purified water to slightly below the plate’s surface at 37 °C. A 10 g weight is attached to the shaft rotating at 25 rpm. When started, the shaft descends, applying load and shear force to the ODT sandwiched between the weight and plate. The ODT absorbs the medium via capillary suction, simulating oral conditions. Harada et al. measured the test medium at 15 s and full disintegration by stopping ODT-101 tests and filtering the medium through No. 325 mesh for taste sensor analysis.

Uchida et al. assessed and compared the taste characteristics of 10 formulations (the original manufacturer’s formulation and nine generics) of amlodipine ODTs through sensory panel testing and a disintegration/dissolution test conducted with OD-mate (Model IMC-14D1, Higuchi Inc., Tokyo, Japan), as depicted in Figure 7, capable of assessing ODT disintegration and subsequent drug release in the oral cavity for up to 60 s. As shown in the figure, the time until the ODT disintegrated completely by the penetrated test medium was measured. Just after disintegration of ODT finished, 20 mL of the test medium containing crushed ODT residue was collected and used for taste sensor measurement. 

A strong correlation between the disintegration times of the 10 amlodipine ODTs estimated in human taste testing and those determined using the OD-mate was established [109]. The effectiveness of the Insent taste sensor in conjunction with OD-mate was also showcased. Consequently, the authors and their co-workers concluded that the combinatory use of OD-mate and the taste sensor is useful for predicting the disintegration/dissolution and assessing the time-course bitterness intensity of amlodipine ODTs in the oral cavity. Furthermore, another study investigated the usefulness of OD-mate in the evaluation of detailed disintegration properties of ODTs [110].

Haraguchi et al. raised concerns about the risk of crushed Vesicare orally disintegrating tablets (ODTs) causing severe bitterness in patients. Following a brief dissolution test of up to 1 min, the Insent taste sensor (AC0), Astree system, and HPLC method were employed to evaluate the time course of bitterness in the oral cavity. Additionally, optical microscopy photography revealed that strongly crushed ODTs led to the complete breakage of functional particles containing API, resulting in the immediate release of the API and the subsequent appearance of severe bitterness [111]. Based on these experimental results, it was consistently noted that bitterness-sensitive sensors such as AN0 or BT0 were always utilized in the evaluation of bitterness scores by the taste sensor.

### 3.2. Assessment of Taste Masking in ODTs by Astree Taste Sensor 

The α-Astree system, an e-tongue, has been utilized for assessing palatability, particularly bitterness. Pattern analysis, typically employing principal component analysis and Euclidean distance between each sample, serves as a key criterion in such evaluations. Here are some examples of experimental data: Nakamura et al. demonstrated α-Astree e-tongue accurately predicted the taste perception of famotidine ODTs and amlodipine ODTs with masked bitterness. The study also assessed its applicability in formulating these tablets. ODTs containing famotidine and amlodipine, known for their bitter taste, were masked using physical (granules coated with ethyl cellulose) or organoleptic (sweetener and flavor) methods, alone or combined. Human taste tests evaluated the effectiveness of these methods. Volunteers rated overall palatability using a 100 mm visual analog scale (VAS). The electronic system was assessed using Euclidean distance and partial least squares (PLS) regression analysis, showing a strong correlation with human VAS scores [112]. 

Kim et al. prepared ODTs involving particles made from ion exchange resin drug complex (IRDC) loaded with donepezil hydrochloride at three different ratios (1:2, 1:1, 2:1) using a spray-drying method. The in vitro taste-masking efficiency was assessed by a “bitterness index (BI)” to link the in vitro Astree sensor and in-human sensory scores [113]. Pattern analysis using the e-tongue was successfully employed to optimize the drug ratio in the preformulation stage. 

Amelian et al. prepared ODTs and lyophilizates with cetirizine dihydrochloride microparticles disintegrated or dissolved in the buccal cavity within seconds without the necessity of drinking. Characterizations such as uniformity of weight and thickness, short disintegration time, and the uniform content of the drug substance were performed. Taste-masking assessments performed by three independent methods (e-tongue evaluation, human test panel, and in vitro drug release) revealed that microparticles with Eudragit^®^ E PO are effective taste-masking carriers of cetirizine dihydrochloride and might be used to formulate orally disintegrating tablets and oral lyophilizates [114].

### 3.3. Assessment of ODFs and Minitablets by Taste Sensor

Changes in European guidance in 2007 seem to encourage the development of dosage forms that are easier to administer to patients of all ages, including children. ODFs and minitablets have been developed, and related research has been performed as follows: The study conducted by Viviane Klingmann et al. compared the acceptability and usability of ODFs with traditional syrup formulations in neonates and infants through a randomized controlled trial [44]. Their findings indicated the following two merits: Firstly, the study demonstrated that the acceptability of ODFs was non-inferior to syrup among neonates and infants. Moreover, ODFs exhibited significant superiority over syrup (*p* < 0.0001), suggesting that they were more acceptable to this patient population. Additionally, the study found that ODFs had significantly superior swallowability compared to syrup (*p* < 0.0001). This indicates that ODFs were easier for neonates and infants to swallow than traditional syrup formulations. These results suggest that ODFs are a safe and effective alternative to liquid formulations for pediatric patients, including very young children. By offering improved acceptability and swallowability, ODFs can contribute to better medication administration and adherence in this vulnerable population. This research is not a sensory study, but the swallowability evaluation seems important. 

Takeuchi et al. introduced an innovative approach employing an Insent taste-sensing system to evaluate the time-evolving bitterness alterations in an orally disintegrating film (ODF) containing either quinine hydrochloride or donepezil hydrochloride [45]. In this technique, a solid film sample is immersed in the test medium with stirring, and the sensor output is recorded. Initially, the sensor output is subdued, followed by a gradual rise over time. This initial suppression indicates efficient taste masking in the oral cavity, while the subsequent increase indicates effective drug absorption, emphasizing the dual efficacy of the film in taste masking and drug absorption.

The ODF first disintegrates in the oral cavity within 180 s, and the substances/particles are swallowed with the saliva directly afterwards. The dissolution/release of the drug then takes place in the stomach or GIT, as described in the European Pharmacopoeia 10th Edition (Ph. Eur. 10) stipulation (Chapter 2.9.1) [115].

In preformulation studies, Birer et al. demonstrated the formulation of an electrospun ODF containing telmisartan, L-arginine, and polyvinylpyrrolidone K90 (PVP) and conducted characterization studies [116]. The release rate of this formulation was found to be much faster than that of the marketed product. Therefore, one of the merits of such an ODF is the enhancement of drug solubility, thereby mitigating the risk of reduced drug absorption. As previously mentioned, if a medication has a bitter taste, achieving taste masking becomes crucial for ensuring both bioavailability and patient adherence.

Hu J et al. conducted a challenging study involving the preparation of orodispersible minitablets (ODMTs) containing carbamazepine (CBZ) as a model drug. These minitablets were fabricated using a 3D-printing technique via semi-solid extrusion. Additionally, the bitterness of the ODMTs was quantitatively assessed using a taste sensor, specifically the TS-5000Z from Insent Co. Ltd. (Atsugi, Kanagawa, Japan) that employs three sensors [117].

The application of an e-tongue to study the release of valsartan from both uncoated and Eudragit E-coated minitablets was investigated by Wesoły [118]. The drug dissolution profiles obtained with ion-sensitive electrodes were compared with standard dissolution tests conducted using the paddle method. The authors discussed the effect of temperature on the correlation between e-tongue measurements and dissolution results, as well as how the pH and ionic strength of the dissolution medium influenced the degradation or release characteristics of the minitablets.

Ekweremadu et al. formulated minitablets tailored specifically for felines, containing amlodipine besylate, integrating an appropriate flavoring agent, and reducing the dosage form size to 2 mm. The choice of L-Lysine as the flavoring agent was chosen due to the dietary and taste preferences of cats. The influence of L-Lysine on the taste perception of the formulation was assessed using a biosensor system (e-tongue) equipped with sensors responsive to bitter tastes. It was suggested that L-Lysine effectively masked the bitterness of the minitablets and did not have an effect on drug dissolution [119]. 

### 3.4. Performance Qualifications of Taste Sensors 

Recent advancements in taste sensor technology have led to remarkable progress, with applications expanding rapidly. It is essential to provide guidance that outlines the usability of sensors in fixed formulation conditions and determines how the extent of taste masking will be assessed. From a regulatory standpoint, studies regarding the performance qualification of commercial taste sensors are desirable in establishing fixed formulation conditions and, particularly, for determining how the extent of taste masking will be determined. 

Woertz et al. conducted a performance qualification regarding the Insent taste sensor SA402B (Atsugi-chi, Japan), which includes seven lipid membranes representing five tastes: bitterness, umami, saltiness, sourness, and astringency. They assessed the specificity, linearity, range, accuracy, precision, detection limits, and quantitation limits for each sensor type, following the guidelines in ICH Q2 (R1). Quinine hydrochloride, a well-known bitter compound, was used as the model substance. The study found a wide linear range (0.01–100 mM) with high precision (RSD < 4%) for most sensors. Interestingly, one sensor demonstrated a lower detection limit (0.0025 mM) for quinine hydrochloride compared to typical human perception [120]. This qualification study also offers valuable insights into the performance and reliability of commercial taste sensors, contributing to the standardization of methodologies for taste assessment in pharmaceutical formulations. 

Woertz et al. compared these systems in sensor technology and performance across software handling, sensors, and measurement procedures [121]. Tests using substances varying in ionic character (sodium saccharin, acetaminophen, ibuprofen, quinine, caffeine) showed both systems detect ionic substances better than neutral ones. TS-5000Z underwent performance qualification; both systems correlated well with human taste assessment, showing reproducible results surpassing panel evaluations. Both systems evaluated the taste masking of ibuprofen and quinine hydrochloride by maltodextrin. The Astree sensor requires data normalization for inter-day comparisons, while Insent’s referencing to a standard solution yields better inter-day consistency. Both systems are valuable for developing taste-masked pharmaceuticals.

Woertz et al. used TS-5000Z (Insent Inc., Atsugi-Chi, Japan) in the verification of the taste masking of bitter quinine hydrochloride in a liquid formulation [122]. They assessed nine types of sweetening agents, along with strong and weak cation ion exchange (IE) resins (Amberlite™ IRP69, Amberlite™ IRP88, and Indion234), and soluble complexing agents (α-, β-, hydroxypropyl-β-, sulfobutyl ether-β-, and γ-cyclodextrin, and maltodextrin). Among these, Amberlite™ IRP88 demonstrated the highest binding capacity for quinine (1.9 g quinine/1 g IE). The inclusion of sulfobutyl ether-β-cyclodextrin (SBE-β-CD) notably diminished the bitter taste of quinine hydrochloride (79% reduction of free quinine hydrochloride). Additionally, the SBE-β-CD formulation was augmented by the incorporation of sodium saccharin as a secondary taste-masking agent. It was also shown that the presence of strawberry flavor and the preservative domiphen bromide did not influence the assessment of taste-masking efficiency. A stepwise approach proved effective for the systematic development of novel taste-masked formulations. 

## 4. Taste Sensor Assessment in Medicinal Plants and Chinese (Herbal) Medicines

It is fascinating to explore the application of taste sensors in evaluating medicinal plants and Chinese medicines, especially considering their potential benefits for conditions like diabetes mellitus and hypertension [123,124]. These natural remedies often contain various compounds that contribute to bitterness or astringency. Long-term oral supplementation with medicinal plants or Chinese medicines may pose adherence challenges due to their bitter or astringent taste. Therefore, assessing the taste, particularly bitterness or astringency, of these substances using taste-sensing systems is crucial. Despite the importance of this area, there have been relatively few studies utilizing e-tongues to assess medicinal plants or Chinese medicines. To my knowledge, the author’s group was among the first to assess the taste of medicinal plants and Chinese medicines using the Insent multichannel sensor. Kataoka’s early publications in the 2000s represent pioneering work in this field [125,126].

Expanding research in this area could provide valuable insights into the taste profiles of medicinal plants and Chinese medicines, potentially informing formulation strategies to improve palatability and adherence. Additionally, such studies may contribute to a deeper understanding of the relationship between taste perception and therapeutic effects in traditional medicine. 

Kataoka et al. conducted numerous studies assessing the taste of various bottled nutritive drinks and Chinese medicines using taste sensors, notably the Insent multichannel sensor [125,126]. In one investigation, Kataoka assessed the taste of 20 commercially available bottled nutritive drinks, employing both human gustatory sensation tests and the Insent multichannel taste sensor [125]. They discovered a positive linear correlation between the intensities of sourness and bitterness as determined by human volunteers and those predicted by the taste sensor (with correlation coefficients of 0.85 and 0.71, respectively). Additionally, the sensor output was able to predict the pungency intensity observed in gustatory sensation tests, with a correlation coefficient of 0.84. These findings suggest that taste sensors could be valuable tools for evaluating the palatability of bottled nutritive drinks. Furthermore, Kataoka et al. assessed the bitterness and astringency of Chinese medicines using quinine solution and tannic acid solution as benchmarks, respectively. They successfully assessed the taste of each Chinese medicine using Euclidean distance analysis with the Insent multichannel sensor [126]. This method allowed for the quantitative assessment of bitterness and astringency, providing valuable insights into the sensory characteristics of Chinese herbal medicines. Overall, Kataoka’s studies demonstrate the utility of Insent multichannel taste sensors in evaluating the taste properties of medicinal plants and Chinese medicines, highlighting their potential as valuable tools in sensory analysis and product development.

In 2016, Li et al. optimized and validated a method using the Astree system as an e-tongue for analyzing the bitterness of traditional Chinese medicines (TCMs) [127]. They utilized this method to evaluate 35 different Chinese medicine formulations, employing diverse analytical approaches, including least squares support vector machine, support vector machine, discriminant analysis (DA), and partial least squares (PLS). This method highlights the potential usefulness of e-tongue technology in evaluating the sensory characteristics of food and beverages.

In the domain of Ayurvedic medicinal plants, Jayasundar et al. conducted sensory and chemical analyses to comprehensively evaluate the “rasa” (taste) of medicinal plants [128]. They also conducted a chemosensory-based Ayurvedic classification of medicinal plants, illustrating how an e-tongue coupled with multivariate statistical analysis can be employed to profile these plants [129]. This emphasizes the adaptability of e-tongue technology in characterizing the taste attributes of medicinal plants and further highlights its potential for utilization in traditional medicine systems such as Ayurveda. Kumar et al. optimized the use of solvent and standardized sample concentration for studying plants from an Ayurvedic perspective of “rasa” (taste) [130]. They determined that Milli-Q water (Merck Millipore, Rahway, NJ, USA) and double-distilled water are appropriate solvents for e-tongue studies of medicinal plants. Through optimizing the solvent and sample concentration, they expanded the scope for taste ranking of medicinal plants based on their rasa properties. Additionally, their research underscored the significance of standardizing and optimizing the concentration of samples and taste standards in Ayurvedic rasa-based studies. This research is particularly useful for the quality control of medicinal plants or their extract products. By ensuring standardized methods for studying the taste properties of these plants, researchers and practitioners can better assess their efficacy and quality. This can contribute to the development of reliable Chinese (herbal) medicines and supplements, enhancing their acceptance and effectiveness in traditional and modern healthcare systems.

## 5. Taste Sensor Assessment for Nutritional Supplement Containing BCAA

The preferred method of nutrition, enteral nutrition (EN), is administered through the gut [131], as it is the most natural means of nutrient absorption and is safer than intravenous nutrition, which poses a risk of immediate bloodstream delivery and infection [132]. Elemental diets (EDs) and low-residue diets (LRDs) have been utilized in the therapeutic management of patients with Crohn’s disease. However, poor taste associated with amino acid-based formulas in EDs has been noted [132], and even in LRDs, taste may be suboptimal, emphasizing the importance of patient education for adherence [133,134].

Aminoleban EN Powder Mix (Otsuka Co. Ltd. Tokyo, Japan) is an orally administered hepatic nutritional supplement designed for patients with liver failure. This formulation includes branched-chain amino acids (BCAAs), including L-Leucine (L-Lue), L-Isoleucine (L-ILeu), and L-Valine (L-Val), which are indispensable amino acids for humans. Notably, L-Lue has been shown to boost protein synthesis by activating the mammalian target of rapamycin (mTOR) signaling pathway in skeletal muscle, adipose tissue, and placental cells [35]. The bitterness observed in Aminoleban EN Powder Mix, an oral nutritional supplement for liver failure, has been attributed to the presence of BCAAs [135], particularly L-Lue, L-ILeu, and L-Val. Hence, our initial objective was to assess the intensity of bitterness associated with various amino acids, including BCAAs, through taste sensor measurements. 

Miyanaga et al. also conducted a pioneering study examining the bitterness of L-Lue, L-ILeu, and L-Val, both individually and in combination, using both taste sensor analysis and gustatory sensation tests [135]. They observed a significant correlation between the results of the human gustatory sensation tests and the sensor output (expressed as R; relative value). However, the component principal analysis (CPA) yielded insignificant results due to the small values obtained. Despite lacking an aromatic ring within their molecular structure, amino acids such as BCAAs are associated with other functional groups like CH_3_, which confer hydrophobic properties and contribute to their bitterness. This finding highlights the complex relationship between the chemical structure of amino acids and their perceived taste characteristics. 

Mukai et al. performed a quantitative assessment of the taste profiles of various total enteral nutrients available in Japan, employing both human gustatory sensation tests and a multichannel artificial taste sensor [136]. In the human gustatory sensation tests, they assessed four primary taste intensities (sweetness, saltiness, sourness, and bitterness), along with 15 palatability scales, using the semantic differential (SD) method. Among these palatability factors, “difficult to drink/easy to drink” was chosen as an overarching measure of palatability due to its significant factor loading in factor analysis. Their results demonstrated a strong positive correlation between overall palatability and sweetness and sourness, while a negative correlation was observed with bitterness and saltiness. Additionally, the incorporation of flavored powder into the amino acid-based enteral nutrient Aminoleban EN Powder Mix significantly improved its palatability. This enhancement was attributed to the sour components of the flavor, such as citric acid, which effectively mitigated the bitterness intensity of the branched-chain amino acids present in the product. Moreover, the sweetness and sourness intensities predicted by the taste sensor showed a high correlation with the outcomes obtained from the human gustatory sensation tests, as illustrated in Figure 8 [136]. This underscores the reliability and validity of the taste sensor in assessing the taste properties of enteral nutrients and highlights its potential as a valuable tool in product development and quality control processes. Regarding the suppression of bitterness in BCAAs, the presence of citric acid in flavored powder containing organic acids proved to be effective in the study. 

Even though a multi-million-dollar industry of nutritional supplements has grown around the concept that dietary supplements of BCAAs alone produce an anabolic response in humans driven by a stimulation of muscle protein synthesis, Wolfe concluded in the review that the claim that the consumption of dietary BCAAs stimulates muscle protein synthesis or produces an anabolic response in human subjects is unwarranted [137].

Nevertheless, many patients suffering from chronic liver failure take the branched amino acid-based enteral nutrient Aminoleban EN Powder Mix every day. Some patients use the medicine by cooling it somehow and diluting it with much water to reduce the bitterness due to BCAAs. The attached flavored powder seems useful for reducing bitterness. From the above, the bitterness intensity could be reduced by the addition of various flavored powders, which could help to maintain patient adherence.

## 6. Taste Sensor Assessment in Bitterness Suppression

In the context of bitterness suppression for target drugs, conventional methods involve physical masking through coating, chemical masking using cyclodextrins, and gustatory masking with sweeteners, as discussed in the introduction.

The use of cyclodextrin is a representative taste-masking method. Adamkiewicz L and Szeleszczuk reviewed applications of cyclodextrins as taste-masking excipients for pharmaceutical purposes, the methods of evaluation of the taste-masking properties, and the factors affecting the outcomes, such as the choice of the proper cyclodextrin or guest–host molar ratio. The conclusions of this review reveal that the application of CDs is not straightforward; nevertheless, this solution can be an effective, safe, and inexpensive method of taste masking for pharmaceutical purposes [138].

Preis et al. prepared a pre-formed resinate of cetirizine HCl and various cyclodextrins, which can be successfully incorporated into the Zydis^®^ freeze-dried formulation. A chemically stable product with a good release profile was obtained in the case of cyclodextrin. This study has also demonstrated that the Insent^®^ taste sensing system is a useful technique for predicting the taste-masking potential of Zydis^®^ formulations. In addition, the taste evaluation results showed that an effectively taste-masked formulation has been achieved using β-cyclodextrin and a cherry/sucralose flavor system, with over 80% of volunteers finding the tablet to be acceptable [139]. Cyclodextrin is safe and permitted as an oral excipient.

Shiraishi et al. assessed the bitter taste-masking effect of chlorogenic acid (CGA) on various pharmaceuticals and elucidated its mechanism using the Insent taste sensor and the surface plasmon resonance (SPR) method [60]. CGA is safe and non-toxic. The bitter-tasting drugs examined included amlodipine besylate (AMD), an antihypertensive drug; diphenhydramine hydrochloride (DPH), an antiallergic drug; donepezil hydrochloride (DNP), used for Alzheimer’s disease; rebamipide (RBM), a gastric ulcer drug; and anti-inflammatory drugs such as etodolac (ETD) and diclofenac sodium (DCF). The sensor output of each drug and the suppression of sensor output by the addition of CGA were assessed. The basic bitter membrane AN0 sensor output of AMD, DPH, and DNP, as well as the astringent taste membrane AE1 sensor output of RBM, ETD, and DCF, increased in a concentration-dependent manner. However, the addition of CGA led to a concentration-dependent suppression of each sensor output. This suggests that CGA effectively masks the bitter taste of these drugs, offering potential applications in improving patient compliance and overall medication experience. Thus, it was anticipated that chlorogenic acid (CGA) would effectively suppress the bitterness of drugs. The degree of sensor output suppression by CGA was found to be significant, with the order of suppression rates being DPH > DNP > AMD ≈ DCF ≈ RBM ≈ ETD. To further investigate the mechanism underlying the bitterness suppression by CGA, we assessed the interaction between CGA and each drug using surface plasmon resonance (SPR). Remarkably, a significant correlation was observed between the binding rate constant (ka) of CGA with each drug and the inhibition rate of sensor output, as shown in Figure 9. This suggests that the combination of taste sensor and SPR techniques enables the estimation of the bitter taste-suppressing effect and elucidation of its mechanism through the analysis of molecular interactions.

Shiraishi further studied CGA’s bitterness suppression, finding both QNA and CFA derived from CGA could dose-dependently suppress DPH’s bitterness. ^1^H-NMR analysis revealed that QNA and CGA’s carboxyl group interacted with DPH’s amine group, suggesting direct electrostatic bitterness suppression. CGA and QNA could serve as effective bitterness-masking agents for DPH, potentially improving medication palatability and adherence [60]. The study by Shiraishi et al. differs from the chemical masking method used with cyclodextrin. It is likely that the interaction between CGA and the drug decreases the free drug fraction or reduces the drug’s affinity to the membrane surface. In this sense, the decrease in sensor membrane potential caused by the co-existence of CGA with the medicine appears to reflect a decrease in the free fraction of the drug in sample solutions.

Kim et al. presented an intriguing study on umami–bitter interactions, particularly focusing on the inhibition of bitterness by umami peptides through the human bitter taste receptor [17]. In their research, the authors investigated umami–bitter taste interactions by exposing umami peptides alongside a bitter substance (salicin) in a Ca^2+^-flux signaling assay using cells expressing hTAS2R16. Five representative umami peptides derived from soybean (Glu-Asp, Glu-Glu, Glu-Ser, Asp-Glu-Ser, and Glu-Gly-Ser) were found to significantly reduce the salicin-induced intracellular calcium influx in a time-dependent manner. Through the Ca^2+^-flux signaling assay utilizing mixtures of salicin and umami peptides, it was determined that all five umami peptides suppressed salicin-induced intracellular calcium influx in a noncompetitive manner. These findings suggest that umami peptides may function to alleviate bitter taste by interacting with bitter taste receptors.

Okuno et al. examined the ability of umami dipeptides (Glu-Glu and Asp-Asp) and their constituent amino acids (Glu and Asp) to suppress bitterness using diphenhydramine (DPH) as a model bitter substance. This assessment was carried out through human panel tests and with the Insent taste sensor [61]. Different concentrations (ranging from 0.001 to 5.0 mM) of Glu-Glu, Asp-Asp, Glu, and Asp significantly reduced the taste sensor response of a 0.5 mM DPH solution in a dose-dependent manner. The effectiveness of umami dipeptides and their constituent amino acids in bitterness suppression followed this order: Asp-Asp > Glu-Glu >> Gly-Gly, and Asp > Glu >> Gly (control), respectively. In human panel tests, concentrations (0.5, 1.0, 1.5 mM) of Glu-Glu, Asp-Asp, Glu, and Asp significantly decreased the bitterness intensity of a 0.5 mM DPH solution, although no statistical difference was observed among the four substances, as depicted in Figure 10.

Notably, there was a significant correlation between the taste sensor readings and results of human panel tests. A surface plasmon resonance study employing hTAS2R14 protein and these substances revealed that Glu-Glu, Asp-Asp, Glu, and Asp had a higher affinity for hTAS2R14 protein compared to Gly-Gly or Gly. Additionally, docking simulation studies involving DPH, Glu-Glu, and Asp-Asp with hTAS2R14 suggested that DPH could bind to a site near the binding position of Glu-Glu and Asp-Asp [61]. In conclusion, umami dipeptides Glu-Glu and Asp-Asp, along with their constituent amino acids, efficiently suppress the bitterness of DPH. It is speculated that this bitterness inhibition is caused by the competitive effect of peptides on the bitterness receptor site.

## 7. Correlation Evaluation between Taste Sensor Response and Human Bitter Receptor Response in BitterDB

Bitter tastants activate bitter taste receptors, known as T2Rs, which are seven-transmembrane G protein-coupled receptors (GPCRs). It is understood that there are 25 types of human bitter taste receptors [53,54,55].

Haraguchi et al. investigated the efficacy of an artificial taste sensor in assessing the bitterness of various drugs [59]. This study involved comparing the responses of the Insent taste sensor with documented reactions from human TASTE2 receptors (hTAS2Rs). To conduct this analysis, we selected 22 bitter compounds commonly found in pharmaceuticals in Japan, all of which are known to interact with hTAS2Rs, as shown in Figure 11. Solutions of these compounds at concentrations of 0.01, 0.03, and 0.1 mM were assessed using five different taste sensors (AC0, AN0, BT0, C00, AE1). We then examined the correlations between the physicochemical properties of the compounds and the responses elicited by both the taste sensors and hTAS2Rs.

Our results indicated that diphenidol, haloperidol, diphenhydramine, dextromethorphan, and papaverine, known ligands of hTAS2R 10 and/or hTAS2R14, were predicted to exhibit strong bitterness, even surpassing that of quinine, based on measurements from the taste sensors. Specifically, we observed significant correlations between the responses of taste sensor BT0 and those of hTAS2R14. Furthermore, compounds with high log P values (≧2.73) also showed significant correlations with the responses of hTAS2R14 (** *p* < 0.01, chi-square test).

In conclusion, our study emphasizes that taste sensor BT0 demonstrates high sensitivity to bitterness and shows a significant correlation with hTAS2R14. Therefore, it proves to be a valuable tool for assessing the bitterness of hydrophobic compounds that interact with hTAS2R14 and their respective inhibitors.

## 8. Novel Taste Sensor Utilizing Allostery and Its Application to Non-Charged Substances

As outlined in the introduction, caffeine, a neutral substance, failed to trigger sensor outputs in all sensors despite being a bitter compound and an agonist for hT2R14 [139]. Toko and co-workers devised a novel membrane modification utilizing aromatic carboxylic acids (e.g., hydroxy-, dihydroxy-, and trihydroxybenzoic acids) such as 2,6-dihydroxybenzoic acid (2,6-DHBA) for the sensor measurement of neutral substances like caffeine and related xanthine derivatives [65,66].

The electrical response of this innovative taste sensor to caffeine was notably amplified compared to sensor responses obtained with the previous BT0. Furthermore, the new sensor response increased proportionally with the concentration of caffeine, theophylline, and theobromine. Additionally, the threshold and ascending trend were consistent with those observed in human perception.

The detection mechanism is believed to involve hydrogen bond interactions between caffeine and the hydroxy group of aromatic carboxylic acid, which influences the dissociated state of the carboxyl group related to H+ binding, as illustrated in Figure 12 [65]. This phenomenon represents an allosteric effect, wherein the binding of caffeine to one site of the aromatic carboxylic acid induces H+ binding at another distant site.

Furthermore, NMR studies have elucidated that the nuclear Overhauser effect (NOE) of intermolecular spatial conformation in solution reveals a preference for 2,6-dihydroxybenzoic acid (2,6-DHBA) to interact with caffeine via hydrogen bonding and stacking configuration between aromatic rings. Identifying the binding form of 2,6-DHBA to caffeine has enabled predictions regarding how the two substances interact, as depicted in Figure 13 [67].

The experimental results suggest the potential for objectively assessing bitterness caused by non-charged bitter substances using taste sensors with allosteric mechanisms and also for developing novel chemical sensors utilizing a similar mechanism.

## 9. Conclusions

This article provides an overview of the use of taste sensors in pharmaceutical applications. Currently, taste sensors, such as e-tongues, are instrumental in verifying the effectiveness of taste masking, differentiating between branded and generic medications, and ensuring quality control across various oral formulations.

I summarized the characteristics of two representative e-tongues and introduced many of their applications in taste masking and formulation design regarding many drugs. It is considered desirable to develop patient-friendly medicine by leveraging the characteristics of each taste sensor. It is necessary to continue enhancing the qualification of taste sensors, including new oral formulations, for pediatric and elderly patients.

It is noteworthy that taste sensors can also forecast alterations in bitterness related to oral pharmaceuticals with various foods, beverages, and other oral pharmaceuticals.

In the future, the development and widespread adoption of additional smaller-size taste sensors might be expected.

My interest moving forward is to develop taste sensors that closely mimic human taste perception. While many bitter medications may not target specific bitterness receptors, understanding how anticancer drug candidates or bitter medications interact with various subtypes of human bitterness receptors could facilitate the discovery of agents to suppress bitterness. For instance, caffeine, a well-known bitter substance and agonist of the human bitter taste receptor hTAS2R14, demonstrates the potential of this approach. The development of taste sensors based on allosteric mechanisms is expected to become a breakthrough in taste sensors that more accurately replicate human taste perception.

## Figures and Tables

**Figure 1 sensors-24-04799-f001:**
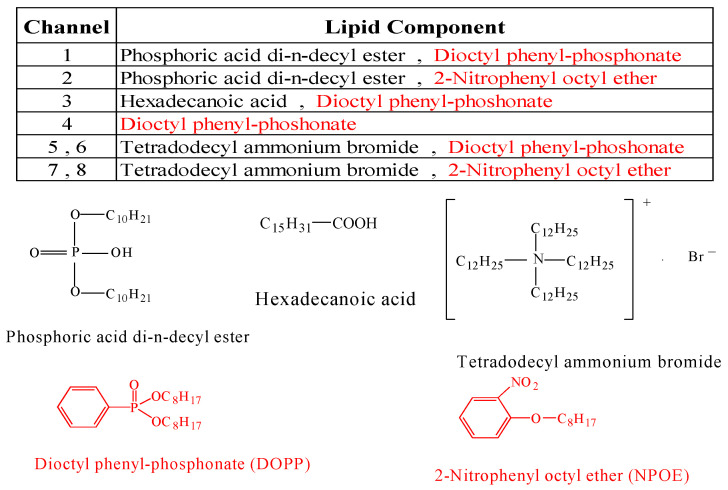
Lipids used for the membranes. Reproduced from [33] with permission from Pharmaceutical Society of Japan. Red-highlighted substances also work as plasticizers.

**Figure 2 sensors-24-04799-f002:**
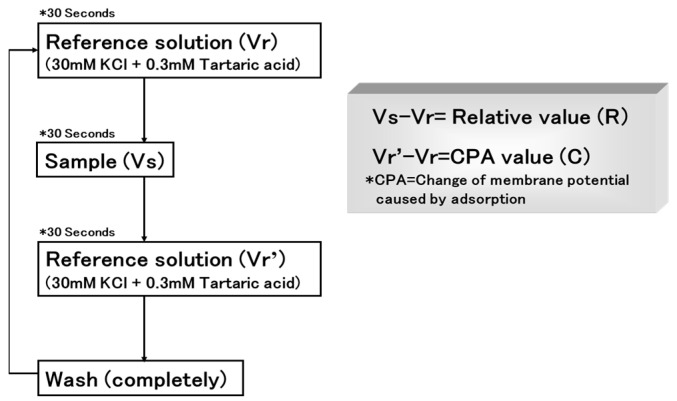
Measuring procedure for taste sensor. Reproduced from ref. [34] with permission from Pharmaceutical Society of Japan. For experiments in ref. [33], Relative value (R) was adopted as an explanatory variable.

**Figure 3 sensors-24-04799-f003:**
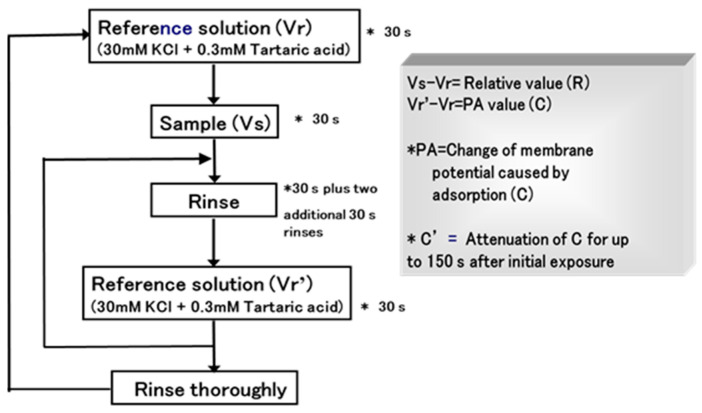
Measuring procedure for taste sensor. Reproduced from [34] with permission from Pharmaceutical Society of Japan.

**Figure 4 sensors-24-04799-f004:**
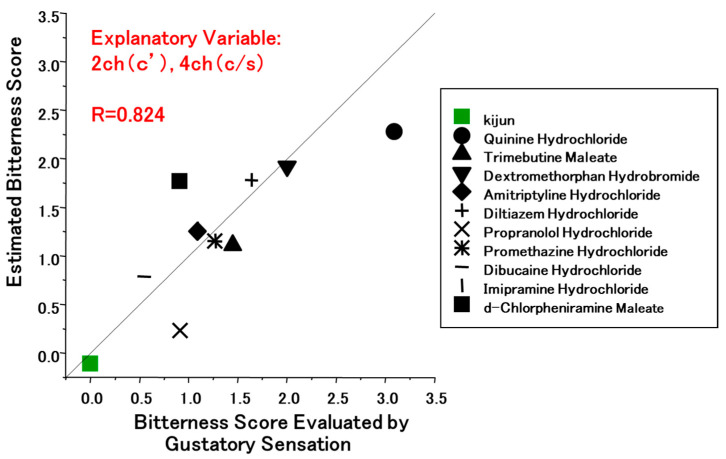
Multiple regression analysis of the data obtained from the taste sensor for 10 medicinal drugs (reproduced from [34] with permission from Pharmaceutical Society of Japan). The vertical axis shows the predicted bitterness score obtained from the taste sensor, while the horizontal axis shows the bitterness score based on human gustatory sensation tests. As explanatory variables, C’ was used for channel 2 and C/S for channel 4.

**Figure 5 sensors-24-04799-f005:**
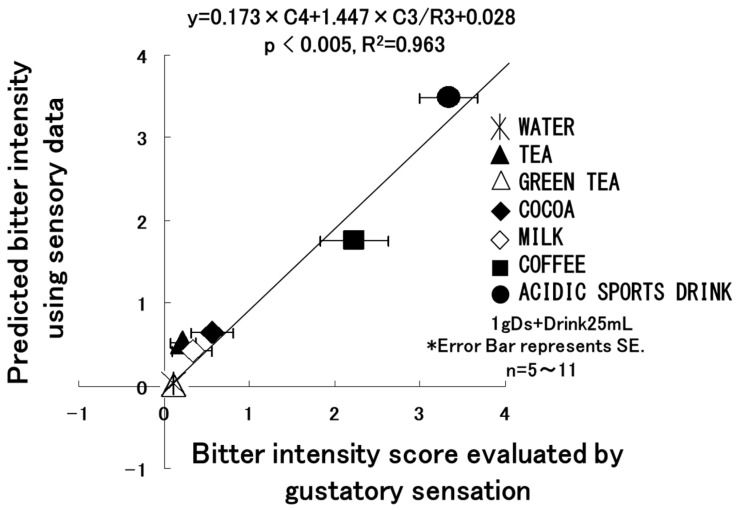
The relationship between the bitterness intensity scores obtained from human volunteers and the predicted values calculated from the equation derived from multiple regression analysis using CPA value from channel 4 and CPA/R value from channel 3 (Reproduced from [36] with permission from Pharmaceutical Society of Japan). Y and X represent the predicted and observed bitterness scores, respectively. For further explanation, see text. Error bars represent the mean plus standard deviation (*n* = 9).

**Figure 6 sensors-24-04799-f006:**
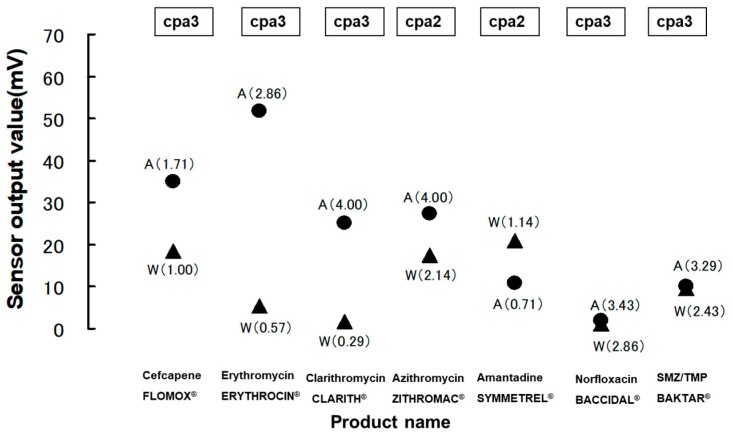
CPA data from channels 2 and 3 of the taste sensor for the seven most bitter drug formulations suspended in water (▲) or acidic sports drink) (●) (reproduced from [37] with permission from Pharmaceutical Society of Japan). Values in parentheses are averages of the bitterness scores obtained in human gustatory sensation tests.

**Figure 7 sensors-24-04799-f007:**
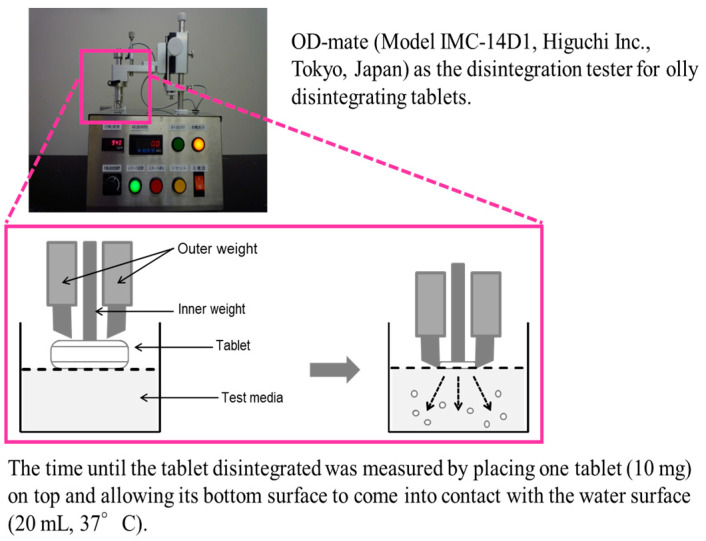
Measurement of disintegration time of famotidine ODT 10 using the OD-mate.

**Figure 8 sensors-24-04799-f008:**
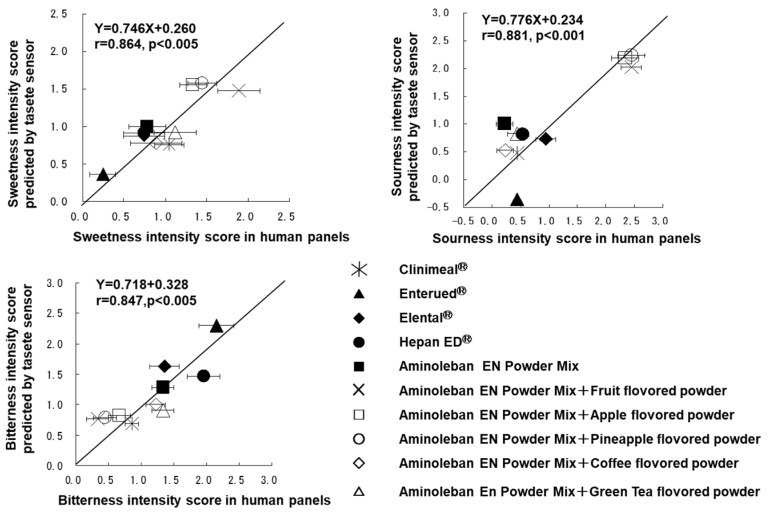
Correlation between the predicted taste intensity by a taste sensor and observed taste intensity by a gustatory sensation test for various enteral nutrients (reproduced from [136] with permission from Pharmaceutical Society of Japan). The data represent the mean of 9 values plus standard errors.

**Figure 9 sensors-24-04799-f009:**
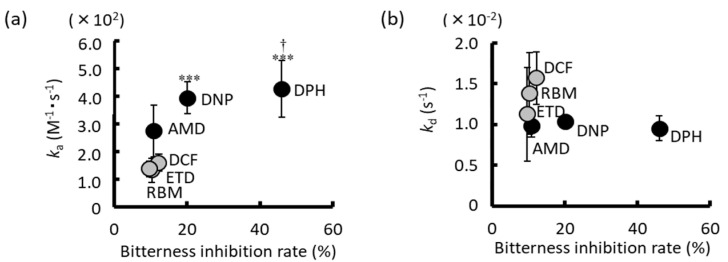
Correlation between the inhibition ratio of taste sensor output and parameters of drug–CGA interaction (reproduced from [60] with permission from Pharmaceutical Society of Japan). (**a**) ka (n = 4, mean ± S.D., *** *p* < 0.001 vs. RBM, † *p* < 0.05 vs. AMD, rs = 0.886, *p* < 0.05, Spearman’s correlation test) and (**b**) kd (n = 4, mean ± S.D. rs = −0.486, N.S., Spearman’s correlation test). Black dots indicate basic drugs (AMD, DNP, DPH), and grey dots indicate acidic drugs (RBM, ETD, DCF).

**Figure 10 sensors-24-04799-f010:**
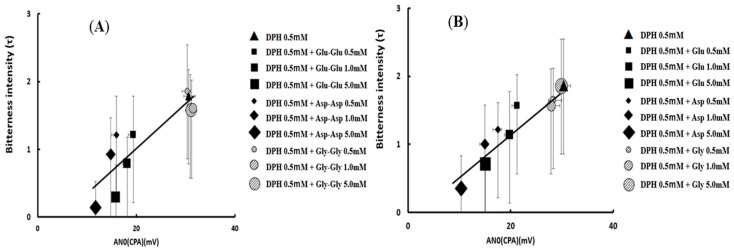
Correlation between taste sensor output (CPA) and bitterness intensity by gustatory sensation test (reproduced from [61] with permission from Pharmaceutical Society of Japan). (**A**) Diphenhydramine hydrochloride (DPH) solutions containing three dipeptide solutions at different concentrations (n = 10, rs = 0.817, *p* < 0.01). (**B**) DPH solutions containing three amino acid solutions at different concentrations (n = 10, rs = 0.970, *p* < 0.001). Both, *p* < 0.001, Spearman’s correlation test.

**Figure 11 sensors-24-04799-f011:**
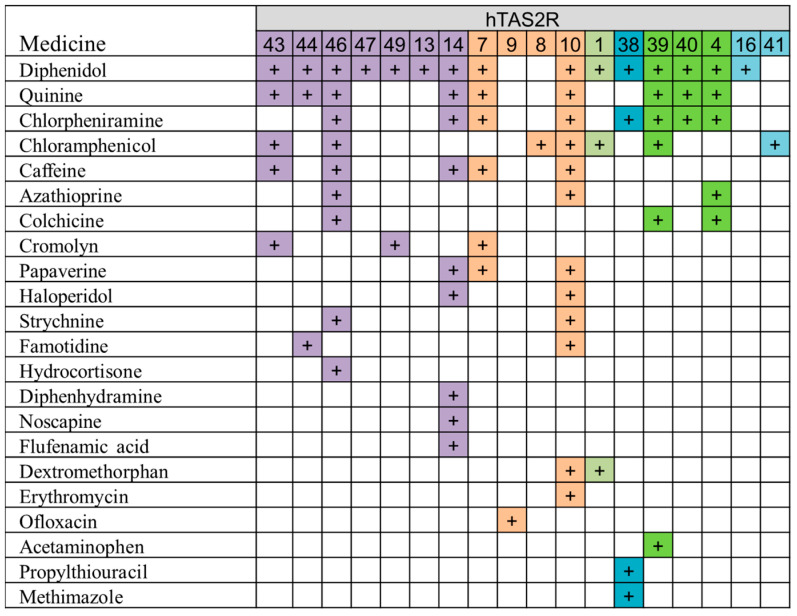
Response profiles of hTAS2Rs stimulated with 22 compounds cited from BitterDB (reproduced from [59] with permission from Pharmaceutical Society of Japan).

**Figure 12 sensors-24-04799-f012:**
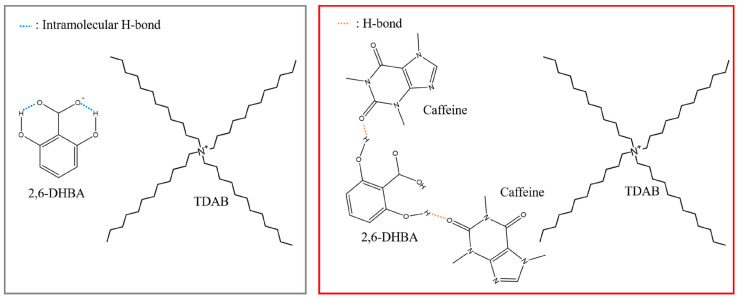
Mechanism of membrane electric potential increase (electric response) when caffeine is measured using the membrane whose surface is modified with 2,6-DHBA. Reproduced from [65].

**Figure 13 sensors-24-04799-f013:**
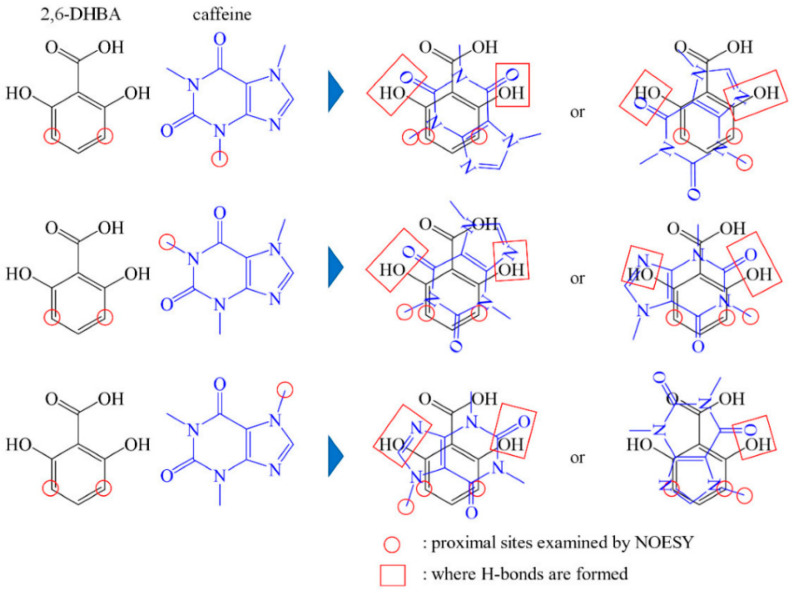
Prediction of the binding form of 2,6-DHBA and caffeine. The binding form was predicted from the information on the proton in proximity, obtained from NOESY. The red circles are the protons in close proximity, obtained from NOESY. The red squares show the positions where H-bonds are formed. Reproduced from [67].

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
