# Peer review of "Taste Sensor Assessment of Bitterness in Medicines: Overview and Recent Topics"

_sensors, 2024, doi:10.3390/s24154799_

Round 1

Reviewer 1 Report

Comments and Suggestions for Authors

The manuscript entitled “Taste sensor assessment of bitterness in medicines: Overview and recent topics” reviews the various applications of taste sensor in the field of studying the bitterness of medicine. In the introduction section, the content is too lengthy, should not be interspersed with too much research from others, is not concise enough, and does not provide a summarized introduction to the main idea of the paper.In the body of the text, it is more of just a recapitulation of what other researchers have done and what they have found, with a lot of repetitive and useless information that is not systematically organized, and there is no specific in-depth reflection and summarization at the end of each section.Conclusion is very general information with poor critical discussion and hence the scientific value is low. The author should catch the point and some content is advised to delete and in order to make this study more accessible to the reader, it is suggested that the entire structure of the paper be reorganized.It is recommended that the author takes the form of a table to summarize the advantages and disadvantages of the application of taste sensor in different studies.

Comment 1: Page 3, Line 135 and Page 5, Line 189  Please standardize the format of references to authors, and suggest that author check and correct the entire text.

Comment 2: Page 4, Line 156  There appears to be a duplication in the format of the references, the author is requested to further confirm this and it is recommended that the full text be checked and corrected.

Comment 3: Page 9, Line 320  In the title "New Generation Formulations", the author needs to consider whether the terminology is correct, and the orally disintegrating tablets and films mentioned later belong to the "dosage form". In addition, the whole chapter 3, from the view of the content of each section, the title grading is unreasonable, and there is a problem of logical confusion, and author should make adjustments.

Comment 4: Page 11, Line 371  Citing another researcher's study should be described using more objective language, such as third perspective, avoiding statements such as "my colleague", and author should correct similar descriptions throughout the text.

Comments on the Quality of English Language

/

Author Response

Thank you for your thoughtful comments concerning my review manuscript entitled “Taste sensor assessment of bitterness in medicines: Overview and recent topics”. I really appreciate the constructive comments and find them extremely beneficial for refining my review manuscript. Firstly, according to your suggestion, the text was dramatically revised thoroughly. Citation of my related references was decreased to 12%, and corresponding Figs and text were deleted. Furthermore each discussion part was tried to enlarge so that the explanation become more scientific, and become better understanding. We are submitting the revised manuscript with the changes as highlighted in red paragraph in the text. The primary revisions to the review manuscript and our responses to the reviewer’s comments are as follows:

Reviewer #1:

  1. General Comment: The manuscript entitled “Taste sensor assessment of bitterness in medicines: Overview and recent topics” reviews the various applications of taste sensor in the field of studying the bitterness of medicine. In the introduction section, the content is too lengthy, should not be interspersed with too much research from others, is not concise enough, and does not provide a summarized introduction to the main idea of the paper.In the body of the text, it is more of just a recapitulation of what other researchers have done and what they have found, with a lot of repetitive and useless information that is not systematically organized, and there is no specific in-depth reflection and summarization at the end of each section. Conclusion is very general information with poor critical discussion and hence the scientific value is low. The author should catch the point and some content is advised to delete and in order to make this study more accessible to the reader, it is suggested that the entire structure of the paper be reorganized.It is recommended that the author takes the form of a table to summarize the advantages and disadvantages of the application of taste sensor in different studies.

Response: Thank you for your suggestion.

Firstly, I reduced self-citation ratio by withdrawing or changing to other author  references regarding following references in original draft, No.5, 28, 29, 48, 53, 55, 58, 60, 67, 93, 94, 95, 96, 114, 117, 122, 128., and corresponding text and Figures. No.60 and 128 were same references, sorry about this. Now my self-citation ratio is 17/140=0.121=12.1%.
In addition, even though not table, advantages and disadvantages of the application of taste sensor in different studies were summarized in new chapter 2.4 Characteristics of Insent Taste Sensor and Astree Sensor System in Application to Pharmaceuticals
According to reviewer’s comment, throughout the text, repetitive and useless information were deleted  from original text, and tried to enlarge discussion part so that the explanation become more scientific, and become better understanding. Finally, I try to make the text more slim so that reader catch the point.

In related to introduction part, I condensed introduction part to70% of original manuscript as follows:
*Lines29-32(Consequently…incombination [12,13] )in original draft was moved to chapter 2 in revised text (Page6 Line200-202).
*Lines 42-51 was deleted including references.
*Lines 66-71(The study….reproducibility) in original draft was also moved to chapter 2 in revised text. revised text (Page3 Line134-139).
*Lines 72-81 was condensed and several my references were deleted.
*Linenes119-126 in original text was completely deleted and corresponding references.

  1. Comment 1: Page 3. Line 135 and Page 5, Line 189 Please standardize the format of references to authors, and suggest that author check and correct the entire text.

Response Thank you for your suggestion. Toko and colleagues (Page 3. Line 135) was changed to Toko and co-workers in revised text (Page 3. Line 103)
And,  Uchida, Kobayashi, et al. (Page 5, Line 189) was changed to Uchida et al. in reivised text  (Page 4. Line 164).

  1. Comment 2: Page 4, Line 156 There appears to be a duplication in the format of the references, the author is requested to further confirm this and it is recommended that the full text be checked and corrected.

Response Thank you for your suggestion.  We deleted (Toko, 1998) (Ikezaki et al., 2001)
(Tahara and Toko., 2013) from text. (Page3. Line 119 in revised text)

  1. Comment 3: Page 9, Line 320 In the title "New Generation Formulations", the author needs to consider whether the terminology is correct, and the orally disintegrating tablets and films mentioned later belong to the "dosage form". In addition, the whole chapter 3, from the view of the content of each section, the title grading is unreasonable, and there is a problem of logical confusion, and author should make adjustments.

Response  I understand the point of your suggestion. According to your suggestion, we use "New Dosage Forms " instead of "New Generation Formulations" in revised text (Page 11. Line 429)

  1. Comment 4: Page 11, Line 371 Citing another researcher's study should be described using more objective language, such as third perspective, avoiding statements such as "my colleague", and author should correct similar descriptions throughout the text.

Response Thank you for your suggestion.  Haraguchi, my colleague in our group (Page 11. Line 371) was changed to Haraguchi et al. (Page 12. Line 494 )  In additiion, all simaltaneous descriptions were rewrote throughtout text.

To Editor

  1. Comment: Thank you for your suggestion. Firstly, I reduced self-citation ratio by withdrawing or changing to other author references regarding following references in original draft, No.5, 28, 29, 48, 53, 55, 58, 60, 67, 93, 94, 95, 96, 114, 117, 122, 128., and corresponding text and Figures. No.60 and 128 were same references, sorry about this. Now my self-citation ratio is 17/140=0.121=12.1%

I tried our best to improve the manuscript and made major changes in the manuscript. I appreciate for Editors/Reviewers’ warm work earnestly and hope that the correction will meet with approval.

Sincerely,

Once again, thank you very much for your comments and suggestions.

June 27th
Takahiro Uchida, Ph.D
Food and Health Innovation Center, Nakamura Gakuen University, 5-7-1, Befu, Jonan-ku, Fukuoka 814-0198, Japan  ttakahiro@nakamura-u.ac.jp

Reviewer 2 Report

Comments and Suggestions for Authors

The article handed in by Uchida regarding taste sensor assessment of bitterness in medication is an interesting overview but also has major weaknesses in the actual version. Please find my comments below:

General remarks:

-        For me the aim of this review is not clear, and a research strategy to clarify which papers were included and why, and which are not included is not given. The article is not an objective review, but summarizes the previous work done by the author and his collogues and gives a short summary of many of these articles which are mainly arranged one below the other. There are no overall discussions or further ideas/comments regarding for example the major findings of these research papers. These are mentioned and shortly summarized in the different sections but not brought into context with each other. Furthermore, approx. 25 % of the listed references of the review are from the author and almost all studies mentioned were performed with one taste sensor system, the system from Insent. This impression is further reassured by the fact that 13 of the 15 figures/charts are reprints from articles published by the author himself. The review has to be generally restructured in order to enable a real overview over all the work done in this field and further studies from other research groups and studies performed with further taste sensors have to be included to meet the expectations a reader has after reading the title of the review.

-        Furthermore, the impression mentioned above you get from reading this review is reinforced by terms like “our colleague”, citation of a reference while using the term “we”, “our group” and are not appropriate in a scientifically sound article. They should be rephrased.

Comments:

-        Abbreviations are not used uniformly: e.g. electronic tongue, e-tongue and E-tongue. Please adapt this.

-        Line 156: should these references also be included here? If yes, then please add numbers and adapt the citation stile.

-        Line 160f: “In this chapter, the author primarily elaborates on the various advantages of the sensor” – this sentence raises expectations that disadvantages would be discussed later. Please also include disadvantages in your contemplations, as this article should be a review including all aspects of the studies performed before.

-        Chapter 2: In this part the reader gets the impression that manly studies performed by the author are included and discussed. Please include also studies from other groups and discuss the core findings of the different articles in this review.

-        Chart 2: Advantages of the introduced semi-continuous measurement method should be discussed and the time needed for one single measurement with the taste sensor should be given, so the reader gets the impression how long this could take, because this is an important parameter, also when it comes to the measurement of orodispersible forms, as discussed later in the review.

-        Line 209: Who is meant by “Author et al.”?

-        Line 222 till 230: please add the reference study cited here. How was the taste masking of the formulation realized and what are the differences in the formulation? Please add more details.

-        Line 244: a sports drink is mentioned here and later in this paragraph the text referrers to an acidic sports drink. Please clarify if this is the same drink. In Figure 3 the term AQUARIOUS is used. Please clarify if this is the (acidic?) sports drink or if this is something else.

-        Figure 5: Lines 271ff mentioned are three macrolide antibiotic formulations, one with clarithromycin and one with azithromycin. I cannot find these five formulations in the figure mentioned in the text and it is not comprehensible for me. Please comment on this. Furthermore, other products like Cefcapene, Norfloxacin and SMZ/TMP are also shown in the figure but not mentioned at all. If shown, they should also be included in the discussion.

-        Line 292: As this is a review article, data has not be presented, as is has to be in a research paper. But, an appropriate discussion of the mentioned results and shown studies is necessary and has to be added in this article.

-        Line 294: did this study also include a human taste panel?

-        Line 324: “presently, ODTs are designed to incorporate small-size functional particles for taste masking”: it is not clear what is meant by that. ODTs were not designed for taste-masking purposes but to directly disintegrate in the mouth. Taste-masking has to be included in the formulation as soon as the taste of the drug is not appropriate.

-        Line 326: “with the emerge of these new formulations”: but the sensors were not developed due to the development of ODTs, furthermore, there are also further orodispersible forms like granules and films. What about liquid formulations that have been on the marked for a much longer time?

-        Lines 336-341: I do not understand why this study was mentioned here. I cannot see the connection to the sections title “taste masking in ODTs”. Please comment on this.

-        Lines 348ff: please closer describe the measurement set-up. Were the measurements performed during the disintegration of the ODT? How long did one measurement last and how long was the rinsing of the sensor? This information is necessary to understand what was done and what was measured in this study.

-        Lines 405-420: It is not clear what the abbreviation ODF means, does it stand for orodispersible film or the plural orodispersible films. Furthermore, later in the paragraph both ODF and ODFs was used. Please clarify and keep it constant.

-        Line 424: The measurement set-up is not clear here. Were several single measurements performed during the disintegration of the dosage form? Indicating an increase in bitterness over time? How was the rinsing of the sensors performed during the measurements and how long did it take?

-        Line 429: again, ODF(s) was introduced here although it was described before. Please consider introducing an abbreviation once and stick to this abbreviation the whole time.

-        Lines 429-430: This statement is not correct or not specific enough, the ODF first disintegrates in the oral cavity within 30 – 180 seconds and the substances/particles are swallowed with the saliva directly afterwards. The dissolution/release of the drug takes then place in the stomach or GIT. Please clarify and add references here.

-        Line 452: could the word “informed” please be replaced, it is not clear what is mentioned by that.

-        Line 456ff: it is not clear what is meant by this last sentence, could you please clarify.

-        Section 3.4: please clarify if validation is really meant here, because many references discussed in this section afterwards are not focusing on the validation but on the qualification of electronic tongues. Please include references showing the difference between qualification of the sensors and its validation e.g. for electronic tongues.

-        Line 467: Woertz et al. do not focus on the validation of the Insent sensor but on the performance qualification. There is a difference between validation and qualification. Please rewrite this section.

-        Line 494: Woertz et al. discuss the verification in their paper and not the validation. According to the publication, the experiments were performed with a qualified system. Please rewrite this section.

-        Lines 507ff: it not clear why this study is mentioned in this section. Did they perform a validation in this study? How was is performed and what were the results?

-        Line 529: “To my knowledge, author group”, who is meant by this? The author’s group? Please rephrase.

-        Line 531: Please add references.

-        Line 656: drug-drug interactions were chosen, although it is not explained why. It seems that it was only chosen to enable the citation of the studies performed by the author.

-        Line 674: CGA was already introduced above and does not have to be mentioned here again.

-        Line 693: Marvin Sketch is not an analytical method, is it? Please rephrase.

-        Section 6: different findings of the studies shown are mentioned but there is no overall discussion of this, so the author could see a benefit in this review.

-        Line 725: When there are not statistical differences observed, then it is not possible to say that there is a significant decrease, as this requires a test of significance.

-        Lines 728-737: please add a reference.

-        Line 744: Which artificial taste sensor was used?

-        Conclusion: please adjust this section according to the changes made. E.g. validation/qualification/verification of the sensors.

Comments on the Quality of English Language

Moderate editing of English language required, comments also inclueded in the section above.

Author Response

Thank you for your thoughtful comments concerning my review manuscript entitled “Taste sensor assessment of bitterness in medicines: Overview and recent topics”. I really appreciate the constructive comments and find them extremely beneficial for refining my review manuscript. Firstly, according to your suggestion, the text was dramatically revised thoroughly. Citation of my related references was decreased to 12%, and corresponding Figs and text were deleted. Furthermore each discussion part was tried to enlarge so that the explanation become more scientific, and become better understanding. We are submitting the revised manuscript with the changes as highlighted in red paragraph in the text. The primary revisions to the review manuscript and our responses to the reviewer’s comments are as follows:

Reviewer #2:

  1. General remarks: - For me the aim of this review is not clear, and a research strategy to clarify which papers were included and why, and which are not included is not given. The article is not an objective review, but summarizes the previous work done by the author and his collogues and gives a short summary of many of these articles which are mainly arranged one below the other. There are no overall discussions or further ideas/comments regarding for example the major findings of these research papers. These are mentioned and shortly summarized in the different sections but not brought into context with each other. Furthermore, approx. 25 % of the listed references of the review are from the author and almost all studies mentioned were performed with one taste sensor system, the system from Insent. This impression is further reassured by the fact that 13 of the 15 figures/charts are reprints from articles published by the author himself. The review has to be generally restructured in order to enable a real overview over all the work done in this field and further studies from other research groups and studies performed with further taste sensors have to be included to meet the expectations a reader has after reading the title of the review.

Response Thank you for your suggestion.
Firstly, I reduced self-citation ratio by withdrawing or changing to other author references regarding following references in original draft, No.5, 28, 29, 48, 53, 55, 58, 60, 67, 93, 94, 95, 96, 114, 117, 122, 128., and corresponding text and Figures. No.60 and 128 were same references, sorry about this. Now my self-citation ratio is 17/140=0.121=12.1%.

Advantages and disadvantages of the application of taste sensor in different studies were summarized in new chapter 2.4 Characteristics of Insent Taste Sensor and Astree Sensor System in Application to Pharmaceuticals

In addition, according to reviewers comment, throughout the text, repetitive and useless information were deleted from original text, and tried to enlarge discussion part so that the explanation become more scientific, and become better understanding. Finally, I try to make the text more slim so that reader catch the point.

  1. Additional general remarks: - Furthermore, the impression mentioned above you get from reading this review is reinforced by terms like “our colleague”, citation of a reference while using the term “we”, “our group” and are not appropriate in a scientifically sound article. They should be rephrased.

Response :  Thank you for your suggestion.  All changed as described below:
Line 64, Toko and colleagues→ Line 60,  Toko and co-workers (revised)
Line 72 Uchida and colleagues→Line 62, Uchida and co-workers
Line 112 Our colleague, Haraguchi,→ Line 91, Haraguch et al.
Line 64, Toko and colleagues→Line 60, Toko and co-workers
Line 72 Uchida and colleagues→Line 62, Uchida and co-workers
Line 112 Our colleague, Haraguchi,→Line 91,  Haraguch et al.
Line 119  The author → The paragraph was deleted for condensation
Line 122 The author and their colleagues →The paragraph was deleted for condensation
Line 135 Toko and colleagues→Line103, Toko and co-workers
Line 166 Uchida and their colleague →Line140, Uchida and co-workers
Line 268 Ishizaka and collegues →Line 243, Ishizaka and co-workers
Line 340 our colleague Ito→The paragraph was deleted for condensation including references
Line 371 Haraguchi, my colleague in our group→Line 494, Haraguchi et al.,
Line 538   Kataoka, a colleague in our group, et al →Line 652, Kataoka et al
Line 551  Kataoka and colleagues →Line 662, Kataoka et al
Line 661  Shiraishi and our colleagues →Line 784, Shiraishi et al.
Line 699  Kawahara, colleague in our group→Line Kawahara et al. (deleted for condense)
Line 716  Okuno and colleagues →Line 832, Okuno et al.
Line 744  Haraguchi, a colleague in our research group →Line 861, Haraguchi et al.
Line 774 Toko and colleagues→Line 887, Toko and co-workers

  1. Comment 1: Abbreviations are not used uniformly: e.g. electronic tongue, e-tongue and E-tongue. Please adapt this.

Response   Thank you for your suggestion. All electronic tongue and E-tongue were changed to  e-tongue through text.

  1. Comment 2: Line 156: should these references also be included here? If yes, then please add numbers and adapt the citation stile.

Response:  Sorry! That was my mistake.   (Toko, 1998) (Ikezaki et al., 2001) (Tahara and Toko., 2013) was deleted in revised text (Page3. Line 119 in revised text)). Only references number 103,104,105 were remained.

  1. Comment 3  -   Line 160f: “In this chapter, the author primarily elaborates on the various advantages of the sensor” – this sentence raises expectations that disadvantages would be discussed later. Please also include disadvantages in your contemplations, as this article should be a review including all aspects of the studies performed before.

Response Thank you for your suggestion. Characteristics including advantage and challenge was newly summaruzed as 2.4 Characteristics of Insent Taste Sensor and Astree Sensor System in Application to Pharmaceuticals.

  1. Comment 4 -   Chapter 2: In this part the reader gets the impression that manly studies performed by the author are included and discussed.

Response Thank you for your suggestion. We added new references of Astree e-tongue studies and introduced in revised text (page 10 line305-page11, 356)

  1. Comment 5:   Chart 2: Advantages of the introduced semi-continuous measurement method should be discussed and the time needed for one single measurement with the taste sensor should be given, so the reader gets the impression how long this could take, because this is an important parameter, also when it comes to the measurement of orodispersible forms, as discussed later in the review.  In the text,  total measuring one sample for five times finished within 30 mins at maximum, whereas in case of semi-continuous measurement method finish

Response Thank you for your suggestion. Usually, measuring time of membrane potential for one sample is 30secs, then washed 30 secs twice, then move to other sample, this one set was repeated five times usually. Therefore, ordinary measuring time for normal sample  as shown in Chart 1, is within 20 mins throughout all measuring scheme. In case of semi-continuous measurement method measurement is finished within 30mins at maximum. Measurement time was added to revised text (p3, Line 145-146, p4 Line,169-171, respectively).

  1. Comment 6: -    Line 209: Who is meant by “Author et al.”❓

Response Thank you for your suggestion. Author et al was replaced by Uchida et al.(p 6,Line184)

  1. Comment 7: Line 222 till 230: please add the reference study cited here. How was the taste masking of the formulation realized and what are the differences in the formulation? Please add more details.

Response: Thank you for you comment, this case only one branded CAMD was evaluatd and compared with CAM powder suspensions. The bitterness of the dry syrup was estimated to be reduced to about 1% of that of an equivalent powder suspension [35] In other words, nearly 99% of the bitter taste was successfully masked in the dry syrup formulation, as demonstrated not only by human gustatory sensation but also by taste sensor prediction. This paragraph was described in revised text (p6,  Line208-211)

  1. Comment 8: Line 244: a sports drink is mentioned here and later in this paragraph the text referrers to an acidic sports drink. Please clarify if this is the same drink. In Figure 3 the term AQUARIOUS is used. Please clarify if this is the (acidic?) sports drink or if this is something else.

Response Thank you for your nice suggestion. Acidic sports drink is general name, therefore
I replaced AQUARIOUS to acidic sports drink according to your suggestion.(see Figure 3 in page7)

  1. Comment 9: -   Figure 5: Lines 271ff mentioned are three macrolide antibiotic formulations, one with clarithromycin and one with azithromycin. I cannot find these five formulations in the figure mentioned in the text and it is not comprehensible for me. Please comment on this. Furthermore, other products like Cefcapene, Norfloxacin and SMZ/TMP are also shown in the figure but not mentioned at all. If shown, they should also be included in the discussion

Response Thank you for your suggestion. According to your suggestion, I added following paragraph in the revised text (p7 line251-254) In the figure, concerning Norfloxacin and SMZ/TMP, the pH of each medicine with and without acidic sports drink is not significantly different; therefore, sensor output en-hancement might not occur. In contrast, in the case of Cefcapene, a moderate increase in pH with an acidic sports drink led to an enlargement in sensor output

Comment 10: Line 292: As this is a review article, data has not be presented, as is has to be in a research paper. But, an appropriate discussion of the mentioned results and shown studies is necessary and has to be added in this article.

Response Thank you for your suggestion. Fhe following short paragraph was newly added into the text in revised text (page, line) as follows for more explanation.
Since pH of L-carbocysteine alone is almost pH 2.94, addition of L-carbocysteine to azithromycin fine granules become acidic condition and thereby expected to give rise to rapid release of azithromycin as basic drug.(p8, Line282-284)

  1. Comment 11: Line 294: did this study also include a human taste panel?

Response Thank you for your suggestion. Yes, was confirmed by many patients. “patients” was added to revised text(p8, Line 287)

  1. Comment 12: Line 324: “presently, ODTs are designed to incorporate small-size functional particles for taste masking”: it is not clear what is meant by that. ODTs were not designed for taste-masking purposes but to directly disintegrate in the mouth. Taste-masking has to be included in the formulation as soon as the taste of the drug is not appropriate.

Response Thank you for your suggestion. Your suggestion is absolutely correct. Precisely, some ODTs are designed to incorporate small-size functional particles for taste masking. Some was added to text in revised text (p11, line 432). like as follow: Presently, some ODTs are designed to incorporate small-size functional particles for taste masking when involved drug show severe bitterness.

  1. Comment 13:    Line 326: “with the emerge of these new formulations”: but the sensors were not developed due to the development of ODTs, furthermore, there are also further orodispersible forms like granules and films. What about liquid formulations that have been on the marked for a much longer time?

Response Thank you for your suggestion. I exaggerated this sentence. I revised sentences as follows: There has been a development of new taste sensor membrane lineups, as observed in the Insent taste sensor application. Additionally, advanced analytical methods have been adopted in the Astree taste sensor system. These update in taste sensor seem contribute to new formulation design and choose of appropriate additives. Of course taste masking and its assessment of conventional dosage form such as liquid formulation must not be forgotten., in revised text(p11, line 434-439).

Comment 14: Lines 336-341: I do not understand why this study was mentioned here. I cannot see the connection to the sections title “taste masking in ODTs”. Please comment on this. ・・・

Response Thank you for you nice suggestion. According to your opinion, This paragaph was removed from here and move to p6, Line 214-218 in revised text.

Comment 15:   Lines 348ff: please closer describe the measurement set-up. Were the measurements performed during the disintegration of the ODT? How long did one measurement last and how long was the rinsing of the sensor? This information is necessary to understand what was done and what was measured in this study.

Response Thank you for your suggestion. Following paragraph was added to the revised text (P12,Line 469-476) The special measurement method involves placing an ODT sample on a stainless steel porous plate with a weighted shaft capable of vertical and rotational movement. A pump automatically adjusts 450 ml of purified water to slightly below the plate's surface at 37°C. A 10g weight is attached to the shaft rotating at 25 rpm. When started, the shaft descends, applying load and shear force to the ODT sandwiched between the weight and plate. The ODT absorbs the medium via capillary suction, simulating oral conditions. Harada et al. measured the test medium at 15 seconds and full disintegra-tion by stopping ODT-101 tests and filtering the medium through No. 325 mesh for taste sensor analysis.

  1. Comment 16: Lines 405-420: It is not clear what the abbreviation ODF means, does it stand for orodispersible film or the plural orodispersible films. Furthermore, later in the paragraph both ODF and ODFs was used. Please clarify and keep it constant.

Response Thank you for your suggestion. We fix to ODF orodispersible film in the text.

  1. Comment 17: Line 424: The measurement set-up is not clear here. Were several single measurements performed during the disintegration of the dosage form? Indicating an increase in bitterness over time? How was the rinsing of the sensors performed during the measurements and how long did it take?

Response Thank you for your suggestion. Experimental conditions were added to the text (p12, line 484-487) As shown in the Figure, Time until ODT disintegrate completely by penetrated test medium was measured. Just after disintegration of ODT finished, 20 mL of the test medium containing crushed ODT residue was collected and used as taste sensor measurement.

  1. Comment 18: Line 429: again, ODF(s) was introduced here although it was described before. Please consider introducing an abbreviation once and stick to this abbreviation the whole time.

Response Thank you for your suggestion. All this type of dosage form was represented as ODF, orodispersed film through the text.

  1. Comment 19: -   Lines 429-430: This statement is not correct or not specific enough, the ODF first disintegrates in the oral cavity within 30 – 180 seconds and the substances/particles are swallowed with the saliva directly afterwards. The dissolution/release of the drug takes then place in the stomach or GIT. Please clarify and add references here. Research ref

Response Thank you for your suggestion. Sorry misunderstanding:
I rewrite that paragaph as follows: ODF first disintegrates in the oral cavity within 180 seconds and the substanc-es/particles are swallowed with the saliva directly afterwards. The dissolution/release of the drug takes then place in the stomach or GIT as described in the European.(p14,Line 562-565 in revised text)

Comment 20:   -    Line 452: could the word “informed” please be replaced, it is not clear what is mentioned by that.

Response Thank you for your suggestion. “informed” was replaced by chosen” in revised text in p14, line587)

  1. Comment 21: Line 456ff: it is not clear what is meant by this last sentence, could you please clarify.

Response Thank you for your suggestion. Paragraph was changed to as follows: It was suggested that L-lysine effectively masked the bitterness of the minitablets, and did not have an effect on drug dissolution [134].( p14, Line590-591)

Comment 22: - Section 3.4: please clarify if validation is really meant here, because many references discussed in this section afterwards are not focusing on the validation but on the qualification of electronic tongues. Please include references showing the difference between qualification of the sensors and its validation e.g. for electronic tongues.

Response Thank you for your suggestion. I erase word validation from this section (page 14)

  1. Comment 23: -     Line 467: Woertz et al. do not focus on the validation of the Insent sensor but on the performance qualification. There is a difference between validation and qualification. Please rewrite this section.

Response Thank you for your suggestion. I use performance qualification in the text.(p14 Lines 601,610,616)

  1. Comment 24: -   Line 494: Woertz et al. discuss the verification in their paper and not the validation. According to the publication, the experiments were performed with a qualified system. Please rewrite this section.

Response Thank you for your suggestion. I use performace qualification in the text.(p14 Line 602,611,617)

Comment 25:   -    Lines 507ff: it not clear why this study is mentioned in this section. Did they perform a validation in this study? How was is performed and what were the results?

Response Thank you for your suggestion. This study was completely removed from revised text.

  1. Comment 26: - Line 529: “To my knowledge, author group”, who is meant by this? The author’s group?

Response Thank you for your suggestion.  I replaced to author’s group (p15, line 644)

  1. Comment 27: Line 531: Please add references.

Response Thank you for your suggestion. Yes, [126,127] was added in text (p15,Line 647)

  1. Comment 28 Line 656: drug-drug interactions were chosen, although it is not explained why. It seems that it was only chosen to enable the citation of the studies performed by the author.

Response Thank you for your suggestion. Drug-drug interaction part was removed from this chapter for condensation. Instead I added example of taste masking of CyD and added the reason and that pargraph was condensed in reviesd text (see p18, line772-782) The one of reason our group chose CDA is safe and non-toxic as mentioned in p18, Line 790.)

The following discsusion part was added into the revised text: The study by Shiraishi et al. differs from the chemical masking method used with cyclodextrin. It is likely that the interaction between CGA and the drug decreases the free drug fraction or reduces the drug's affinity to the membrane surface. In this sense, the decrease in sensor membrane potential caused by the co-existence of CGA with the medicine appears to reflect a decrease in the free fraction of the drug in sample solutions. (p19, Line 815-819.)

  1. Comment 29: -  Line 674: CGA was already introduced above and does not have to be mentioned here again.

Response Thank you for your suggestion. CGA was deleted from text.

  1. Comment 30: Line 693: Marvin Sketch is not an analytical method, is it? Please rephrase.

Response Thank you for your suggestion. But, we erase following paragraphs from text coz those study was bitterness due to drug-drug interaction, we delete corresponding 4 references.

In addition, our research group has investigated the bitterness inhibition effect of 5’ adenylic-acid (AMP) and the bitter interaction between combined drugs using the Insent taste sensor and other analytical methods, including 1H-NMR spectroscopic analysis and Marvin Sketch [93,94]. The observed bitterness inhibition was attributed to drug-drug in-teractions mediated by factors such as electrical interactions, hydrophobic interactions, or hydrogen bonding. Especially AMP and related nucleic compounds are crucial molecules in biological processes. They serve as building blocks for DNA and RNA, essential for ge-netic information storage and protein synthesis. AMP also plays roles in cellular energy transfer and signaling pathways and permitted as a component of artificial milk. Kawahara et al. evaluated peripheral bitterness inhibition effect of AMP on the tri-methoprim (TMP) and sulfamethoxazole (SMZ) combination formulation based on Insent taste sensor [93]. The taste sensor values of TMP solutions with different concentrations show large sensor output in correlation with the concentration of TMP, whereas no sensor output in shown for the SMZ solutions. Therefore, the bitterness of this combination for-mulation is mainly due to TMP.

  1. Comment 31: -    Section 6: different findings of the studies shown are mentioned but there is no overall discussion of this, so the author could see a benefit in this review.

Response Thank you for your suggestion. As mentioned above, we added taste masking by CyD was introduced inclulding CDA as non-toxic substance.

  1. Comment 32: Line 725: When there are not statistical differences observed, then it is not possible to say that there is a significant decrease, as this requires a test of significance.

Response Thank you for your suggestion. We slightly changed sentence as follows: In human panel tests, regarding concentrations (0.5, 1.0, 1.5 mM) of Glu-Glu, Asp-Asp, Glu, and Asp, were significantly decreased the bitterness intensity of a 0.5 mM DPH solution, although no statistical difference was observed among the four substances as depicted in Figure 8. (p20,Line841-844)

  1. Comment 33: Lines 728-737: please add a reference.

Response Thank you for your suggestion. We added [92] in revised text.

  1. Comment 34:   Line 744: Which artificial taste sensor was used?

Response Thank you for your suggestion.  Insent taste sensor was added to revised text (page 21, line 863)

  1. Comment 35: -   Conclusion: please adjust this section according to the changes made. E.g. validation/qualification/verification of the sensors.

Response Thank you for your suggestion. We removed ‘validation’ from conclusion.

To Editor

  1. Comment: Thank you for your suggestion. Firstly, I reduced self-citation ratio by withdrawing or changing to other author references regarding following references in original draft, No.5, 28, 29, 48, 53, 55, 58, 60, 67, 93, 94, 95, 96, 114, 117, 122, 128., and corresponding text and Figures. No.60 and 128 were same references, sorry about this. Now my self-citation ratio is 17/140=0.121=12.1%

I tried my best to improve the manuscript and made major changes in the manuscript. I really appreciate for Editors/Reviewers’ warm work earnestly and hope that the correction will meet with approval.

Sincerely,

Once again, thank you very much for your comments and suggestions.

June 27th
Takahiro Uchida, Ph.D
Food and Health Innovation Center, Nakamura Gakuen University, 5-7-1, Befu, Jonan-ku, Fukuoka 814-0198, Japan  ttakahiro@nakamura-u.ac.jp

Round 2

Reviewer 2 Report

Comments and Suggestions for Authors

Thank you for the revision.

Comments on the Quality of English Language

Minor changes have to be made.